# Bayesian Cloud Top Phase Determination for Meteosat Second Generation

Johanna Mayer[1], Luca Bugliaro[1], Bernhard Mayer[1,2], Dennis Piontek[1], and Christiane Voigt[1,3]

[1]Deutsches Zentrum für Luft- und Raumfahrt, Institut für Physik der Atmosphäre, Oberpfaffenhofen, Germany
[2]Ludwig Maximilians Universität, Institut für Meteorologie, Munich, Germany
[3]Johannes Gutenberg-Universität, Mainz, Germany

**Correspondence:** Johanna Mayer (johanna.mayer@dlr.de)

**Abstract.** A comprehensive understanding of cloud thermodynamic phase is crucial for assessing the cloud radiative effect and is a prerequisite for remote sensing retrievals of microphysical cloud properties. While previous algorithms mainly detected ice and liquid phases, there is now a growing awareness for the need to further distinguish between warm liquid, supercooled and mixed phase clouds. To address this need, we introduce a novel method named ProPS (PRObabilistic cloud top Phase retrieval for Seviri), which enables cloud detection and determination of cloud top phase using SEVIRI, the geostationary passive imager aboard Meteosat Second Generation. ProPS discriminates between clear sky, optically thin ice (TI), optically thick ice (IC), mixed phase (MP), supercooled liquid (SC), and warm liquid (LQ) clouds. Our method uses a Bayesian approach based on the cloud mask and cloud phase from the lidar-radar cloud product DARDAR. Validation of ProPS using six months of independent DARDAR data shows promising results: The daytime algorithm successfully detects 93% of clouds and 86% of clear sky pixels. In addition, for phase determination, ProPS accurately classifies 91% of IC, 78% of TI, 52% of MP, 58% of SC and 86% of LQ, providing a significant improvement in accurate cloud top phase discrimination compared to traditional retrieval methods.

## 1 Introduction

Understanding and correctly identifying clouds and their thermodynamic phase in satellite remote sensing is crucial for several reasons. First, the phase critically affects cloud-radiation interactions (Choi et al., 2014; Komurcu et al., 2014; Matus and L'Ecuyer, 2017; Forster et al., 2021; Cesana et al., 2022) and numerous studies have demonstrated the influence of the cloud phase on climate sensitivity in general circulation models (Gregory and Morris, 1996; Doutriaux-Boucher and Quaas, 2004; Cesana et al., 2012; Tan et al., 2016; Bock et al., 2020). Furthermore, phase transition processes depend on various factors like temperature, aerosol abundance and type, the Wegener-Bergeron-Findeisen process, vertical velocity, turbulence and are thus difficult to understand and model (Mioche et al., 2015; Korolev et al., 2017; Coopman et al., 2021; Ricaud et al., 2022). Accurate observations of cloud occurrence and their thermodynamic phase are therefore essential to improve their representation in climate models (Atkinson et al., 2013; Cesana et al., 2015; Matus and L'Ecuyer, 2017; Moser et al., 2023; Hahn et al., 2023; Kirschler et al., 2023). Second, the reliable detection of clouds and the determination of their phase is a critical first step for the remote sensing retrieval of cloud properties such as optical thickness, effective particle radius and water path. Ice and liquid

cloud particles have different scattering and absorption properties, and an incorrect phase assignment can lead to significant errors in remotely retrieved cloud properties (Marchant et al., 2016).

  Passive sensors aboard geostationary satellites play an important role in the observation of clouds and their thermodynamic phase. The advantages of these sensors are their wide field of regard and their ability to observe the same area at any time of day, allowing to study the temporal evolution of clouds with high temporal resolution. However, determining the thermodynamic

phase of clouds using passive sensors is a challenging task. In the past, passive sensor phase retrievals often only distinguished between ice and liquid clouds (or ice/liquid/unknown) (e.g., Key and Intrieri, 2000; Knap et al., 2002; Baum et al., 2012; Bessho et al., 2016; Marchant et al., 2016; Platnick et al., 2017; Benas et al., 2017). More recently, retrieval algorithms have been developed for imagers on geostationary satellites like the Advanced Baseline Imager (ABI) aboard GOES-R and the Advanced Himawari Imager (AHI) aboard Himawari-8, allowing for a further distinction between mixed-phase, liquid, and, in

the case of ABI, supercooled liquid cloud tops (Pavolonis, 2010; Wang et al., 2019; Li et al., 2022). Nevertheless, accurately distinguishing between phases beyond just liquid and ice remains challenging (Korolev et al., 2017). Also, Mayer et al. (2023) show that mixed-phase and supercooled cloud tops are often present over the Meteosat disc, not only in regions like the Southern Ocean, and thus deserve dedicated retrieval algorithms.

  We have developed a new method for cloud detection and cloud top phase determination for the Spinning Enhanced Visible

and Infrared Imager (SEVIRI) on board the geostationary Meteosat Second Generation (MSG) satellite (Schmetz et al., 2002) using a Bayesian approach. Our focus is on the identification of mixed phase and supercooled liquid clouds in addition to the 'traditional' purely ice and warm liquid cloud tops. We use the Lidar-Radar cloud product DARDAR (liDAR/raDAR, Delanoë and Hogan, 2010) as the basis for this method. DARDAR is based on the combination of active radar and lidar measurements from the A-Train satellites CloudSat and CALIPSO and provides a consolidated classification of the measured clouds into

different cloud phases. Synergistic lidar-radar techniques are considered the most reliable for cloud phase determination from satellites because the used instruments are complementary due to their different penetration depths and different particle size sensitivities (Wang, 2012; Delanoë and Hogan, 2008; Zhang et al., 2010; Korolev et al., 2017; Ewald et al., 2021). Over the years, they have been widely used to study the global horizontal and vertical distribution of cloud occurrence and cloud phase (Okamoto et al., 2010; Wang, 2012; Mioche et al., 2015; Matus and L'Ecuyer, 2017; Listowski et al., 2019). For our new phase

retrieval method, we use the DARDAR product as ground truth for cloud and phase occurrence which can distinguish between warm liquid, supercooled liquid, mixed phase and ice. We collocate five years of these data with SEVIRI measurements in selected channels and ancillary data to create a large collocated data set with information on the cloud top phase from DAR-DAR. Our method then uses a probabilistic Bayesian approach as follows: We compute a prior representing the probability of cloud and phase occurrence as well as probabilities for SEVIRI channel measurements from the collocated data set. We update

the prior using Bayes' formula with each successive SEVIRI measurement, resulting in a probability for cloud occurrence and for its top phase based on the prior information and the selected SEVIRI measurements. The SEVIRI channels used in this calculation include three infrared channels (centred at $8.7\,\mu$m, $10.8\,\mu$m and $12\,\mu$m), two visible channels ($0.6\,\mu$m and $1.6\,\mu$m) and a local texture parameter derived from the $10.8\,\mu$m channel.

Bayesian approaches have proven successful in various classification problems using satellite data (Merchant et al., 2005; Mackie et al., 2010; Heidinger et al., 2012; Pavolonis et al., 2015; Meirink et al., 2022). One advantage of the Bayesian approach is its ability to handle complexity and consolidate diverse spectral information from different SEVIRI channels into a single metric (Pavolonis et al., 2015). Furthermore, it is straightforward to define a quality parameter for the result, since the outcome of a Bayesian approach is a probability.

To test the performance of our method we validate it using six months of DARDAR data, which were not used for the computation of probabilities in order to keep the validation independent.

## 2  Data set

### 2.1  DARDAR-MASK

As ground truth for cloud occurrence and cloud thermodynamic phase, this study uses the product DARDAR-MASK, part of the synergistic active remote sensing product DARDAR, specifically the *DARMASK_Simplified_Categorization data* set (Delanoë and Hogan, 2010; Ceccaldi et al., 2013). DARDAR-MASK is derived from the sun-synchronous, low-earth orbit satellites CloudSat (Stephens et al., 2002) and CALIPSO (Winker et al., 2003). To distinguish between cloud phases, DARDAR-MASK uses the wet bulb temperature derived from the ECMWF-AUX dataset (Benedetti, 2005) and the extent of cloud layers as well as the different sensitivities of lidar and radar to cloud particles of varying sizes: cloud layers containing water have a strong lidar backscatter and subsequent attenuation; the CloudSat radar is mostly only sensitive to the larger ice crystals (Hogan et al., 2003). DARDAR-MASK provides vertically resolved cloud thermodynamic phase along the track of the CALIPSO and CloudSat satellites with a spatial resolution of 1.1 km along track and 60 m in the vertical direction. For brevity, we use "DARDAR" instead of DARDAR-MASK to describe the cloud product in the following. An example curtain of DARDAR can be seen in the background of Fig. 7. We collocate five years (2013–2017) of DARDAR data with observations of the passive instrument SEVIRI aboard the geostationary satellite Meteosat-9 (part of the Meteosat Second Generation series) by merging overpasses of the polar orbiting satellites with the corresponding SEVIRI pixel for each time and latitude-longitude combination. The collocated DARDAR data are then aggregated to the spatial resolution of the SEVIRI sensor ($3 \times 3$ km$^2$ at the sub-satellite point). Details on how this collocation is done can be found in Mayer et al. (2023). From the DARDAR data we extract two key pieces of information for each SEVIRI pixel: 1) whether a pixel is clear or cloudy, and 2) a cloud top phase. This cloud top phase at SEVIRI resolution is defined by horizontal and vertical averaging of DARDAR gates using a simplified penetration depth (Mayer et al., 2023). We distinguish between warm liquid (LQ), supercooled liquid (SC), mixed phase (MP) and ice. MP cloud tops in SEVIRI resolution are defined as containing either only gates classified as mixed-phase by DARDAR or a mixture of liquid, ice and/or mixed-phase DARDAR gates in the cloud top gates considered for the collocation (see Mayer et al. (2023) for details). To ensure that the averaging over DARDAR gates for a SEVIRI pixel is not done over two different clouds, the gates are all required to have a similar cloud top height. For multilayered clouds, e.g., a high cirrus cloud on top of lower clouds, only the uppermost cloud layer is considered. For pure ice clouds we use information on the optical thickness contained in DARDAR to distinguish further between optically thin ice (TI) and thick ice (IC), where we use

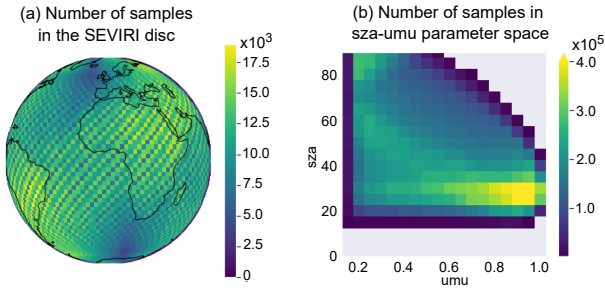

**Figure 1.** (a) Number of samples on the SEVIRI disc in latitude-longitude boxes of $2.5° \times 2.5°$. (b) Number of samples in sza-umu (solar zenith angle - cosine of the satellite zenith angle) parameter space.

an optical thickness $\tau = 2$ as threshold. We employ this distinction since TI and IC have different radiative properties and are typically detected by different channel (combinations) of SEVIRI (see Sect. 4). The optical thickness threshold is consistent with the cloud type categories of GOES-R (Pavolonis, 2010). To combine both aspects (cloudy/clear and cloud top phase), we

introduce a "cloud state parameter", denoted as $q \epsilon \{\mathrm{clear, TI, IC, MP, SC, LQ}\}$. Note that in the following, when we use the terms "cloud state" or "cloud phase" in the context of our retrieval, we are referring to the phase of a cloud only at the cloud top - as passive imagers such as SEVIRI cannot penetrate deep into a cloud.

## 2.2 Distribution of samples

Figure 1(a) shows the distribution of samples in the SEVIRI disc in latitude-longitude boxes of $2.5° \times 2.5°$. The figure demon-

strates the good coverage of samples over the entire SEVIRI disc.

The DARDAR data is obtained from polar orbiting satellites that follow a sun synchronous orbit. Consequently, it can only provide information about clouds during the overflight times. This characteristic of the data has implications for our retrieval process, particularly for the use of solar channels and their dependence on solar and satellite viewing angles. Figure 1(b) shows the distribution of samples in the parameter space spanned by the solar zenith angle (sza) and the cosine of the satellite

zenith angle (umu). Notably, there are two regions in this parameter space where no samples are available: one for sza values below $20°$, and another for combinations of high umu and sza values. The use of solar channels in the retrieval is handled differently for these two regions: For sza values below $20°$, the probabilities employed in the retrieval process are obtained from probabilities for sza values larger than $20°$. For the regions of the parameter space without samples for high sza and umu combinations, the solar channels are effectively not used. In a Bayesian update, this is done by imposing flat probability

distributions for the solar channels in these regions of the parameter space, i.e. the cloud state probabilities are not changed by the solar channels. This is further explained in Sect. 6. In addition, since the DARDAR data do not contain data points at the sunglint, we also impose flat probability distributions for the solar channels close to the sunglint, defined as sunglint angles below $20°$.

There are samples available for all other combinations of umu and sza. However, it is important to note that the data set does not include all of these possible combinations of angles for every latitude. For instance, at low latitudes, the overflight times always occur around noon, resulting in relatively low sza values (between 20° and 40° for latitudes between 0° and $10°N/S$). The statistics for large sza values originate consequently from clouds in higher latitudes. This discrepancy could introduce a bias when using solar channels depending on angles, as meteorological and microphysical conditions in high latitudes may differ from those in lower latitudes.

In addition, as CloudSat operated in daylight-only mode, our data set only includes samples collected during the day. This could potentially introduce a bias in the nighttime retrieval for clouds whose properties differ between night and day.

## 2.3 Ancillary data

In addition, we include ancillary data such as surface temperature and surface type in the collocated data set. The surface temperature data are obtained from the ERA5 reanalysis (Hersbach et al., 2020) and interpolated to the SEVIRI grid. For surface type classification we have adopted the International Geosphere-Biosphere Programme (IGBP) scheme (Loveland and Belward, 1997) provided in the MODIS L3 product MCD12C1 (Friedl et al., 2010). Surface types are grouped into five categories (water, barren, permanent ice and snow, forest and vegetation excluding forest) and projected onto the SEVIRI grid (for details see Strandgren et al., 2017). In summary, our collocated data set includes the cloud state parameter $q$ from DARDAR, SEVIRI observations and ancillary data from ERA5 and IGBP for five years of data. These five years of data amount to over 40 million data points. The use of all these years should ensure that a reasonable amount of annual variability is accounted for.

## 3 Bayes approach applied to satellite data

The output of our new cloud state retrieval method ProPS (PRObabilistic cloud top Phase retrieval for Seviri) is a probability for the cloud state given all (useful) SEVIRI measurements (as defined in Sect. 4) and ancillary data. In the following we explain how this probability is computed with the help of Bayes formula. Figure 2 shows a schematic of the method.

### 3.1 Bayes method

First, we use the collocated data set to compute probabilities $P(q|A)$ for the occurrence of each cloud state $q$ conditioning on a set of ancillary parameters $A$ independent of the satellite observations. These probabilities serve as prior of the cloud state distribution and are updated for each SEVIRI measurement. The updated probability for the cloud state, $P(q|M_1, A)$, given a SEVIRI measurement $M_1$ (i.e. a brightness temperature (BT), a brightness temperature difference (BTD) or a solar observation, see below) and the set of ancillary parameters $A$ already mentioned above is calculated using Bayes formula

$$P(q|M_1, A) = \frac{P(M_1|q, A)P(q|A)}{P(M_1|A)}. \tag{1}$$

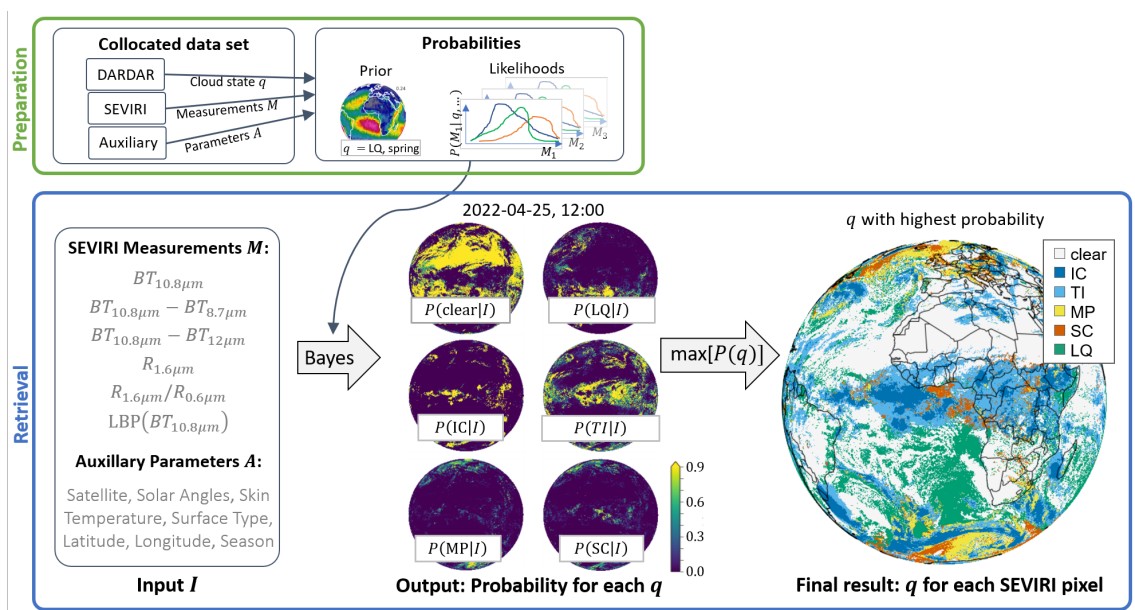

**Figure 2.** Scheme of the phase retrieval method ProPS. The green box shows the preparation for the retrieval, i.e. the calculation of the probabilities from the collocated data set. The blue box shows the phase retrieval steps of ProPS.

The first term in the numerator, $P(M_1|q, A)$, is a conditional probability for the SEVIRI measurement $M_1$ and can be derived from the collocated SEVIRI-DARDAR data set (Sect. 2). The denominator $P(M_1|A)$ acts as a normalisation factor. It can be computed by breaking it down for each possible cloud state $q$, leading to the following decomposition: $P(M_1|A) = \sum_q P(M_1|q, A)P(q|A)$. Note that this is equal to the numerator of Eq. 1 summed over all cloud states $q$. Hence, all terms to compute the updated probability $P(q|M_1, A)$ can be derived from the collocated data set. We repeat the same step for subsequent SEVIRI measurements. Updating the probability with a second SEVIRI measurement $M_2$ leads to

$$P(q|M_2, M_1, A) = \frac{P(M_2|q, M_1, A)P(M_1|q, A)P(q|A)}{P(M_2|M_1, A)P(M_1|A)}, \tag{2}$$

with Bayes' formula being applied twice. For a series of $n$ measurements, the probability for the cloud state $q$ given all the measurements $M := (M_1, M_2, ..., M_n)$ and ancillary parameters $A$ can be expressed as

$$P(q|M, A) = \frac{1}{N} P(M_n|q, M_{n-1}, ..., M_1, A)...$$
$$P(M_2|q, M_1, A)P(M_1|q, A)P(q|A), \tag{3}$$

with the normalization factor

$$N = P(M_n|M_{n-1}, ..., M_1, A)...P(M_2|M_1, A)P(M_1|A). \tag{4}$$

Thanks to equation 3 we can compute a probability for the cloud state $q$ that takes into account (i) prior knowledge about $q$, (ii) all SEVIRI measurements $M$ and (iii) all ancillary parameters $A$.

The data requirements for calculating each probability scale with the number of parameters used as conditions. Fortunately, the conditional probabilities on the right hand side of Eq. 3 can be simplified by considering the dependencies of the different SEVIRI channels. For example, if the measurement of one channel, $M_2$, is (approximately) independent of the measurement of another channel, $M_1$, then its probability reduces to $P(M_2|q, M_1, A) = P(M_2|q, A)$. Similarly, if a measurement is independent of certain auxiliary parameters, these parameters can be removed from the set $A$ in the conditional probability (i.e. $A = \{a_1, a_2, a_3, ...\} \rightarrow A = \{a_1, a_3, ...\}$ if $M_2$ is independent of $a_2$). This simplification step is essential to ensure that the probabilities are meaningful and statistically valid. Given the size of our data set of about 40 million data points, we limit the number of conditions to a maximum of four per probability to ensure statistical validity. In cases where a SEVIRI measurement depends on more than four of the parameters in its conditional probability, we carefully select which of these parameters are the most significant and focus on these, removing the less significant parameters. The selection of channels and conditions for each probability is further explained in the following section (Sect. 4).

## 3.2 Retrieval result

The result of Eq. 3 is a probability for each cloud state $q$. As the final result of the retrieval method, we choose the most likely cloud state, $q^*$, i.e. the cloud state with the highest probability for each SEVIRI pixel

$$q^* = \max_q(P(q|M, A)). \tag{5}$$

Thus, the final result is one cloud state per SEVIRI pixel.

## 3.3 Measure of certainty

There are several advantages of using (Bayesian) probabilities: First, they allow to incorporate prior knowledge. This is in contrast to traditional decision tree models, which typically do not take this valuable information into account. Second, Bayes' formula provides a standardised approach to integrating information from different channel measurements into a single, objective metric. It eliminates the need for arbitrary rules when faced with conflicting cloud state indications from different measurements. Third, the approach maintains transparency; one can clearly understand the origin of the probability values assigned to each cloud state. Finally, since the outcome is a probability for each cloud state, it is straightforward to develop a measure of certainty (a quality measure) associated with the outcome. We define the certainty $c$ as the difference between the probability for $q^*$ and the average probability of the remaining other cloud states $q'$

$$c = P(q^*|M, A) - \frac{1}{5} \sum_{q'} P(q'|M, A). \tag{6}$$

This certainty is a number between zero and one. It is close to one when the highest probability is much larger than the other probabilities. The certainty becomes small when the probabilities for other cloud states are close to the highest probability.

## 4 Selection of channels and dependencies

This section describes which SEVIRI channels and conditions are used for each probability. From the collocated data set we have the following set of ancillary parameters

$$A = \{\mathrm{sza}, \mathrm{umu}, \mathrm{sfc}, \mathrm{skt}, \mathrm{lat}, \mathrm{lon}, \mathrm{season}\}, \tag{7}$$

where sza is the solar zenith angle, umu is the cosine of the satellite zenith angle, sfc is the surface type, skt is the surface temperature, lat is the latitude, lon is the longitude and season is one of the four seasons of the year (December – January – February, March – April – May, June – July – August or September – October – November).

To choose the SEVIRI channels and their most important dependencies for the retrieval, we combine physical/theoretical principles of the physics involved with statistical tools. First, we select channels and channel combinations that are known to carry information about the cloud state. We also consider only a selection of conditions for the probability of each channel (combination) that make sense from a physical perspective. From this selection of physically meaningful conditions, we decide on the optimal conditions for the probability of each channel (combination) using the statistical tool of *mutual information* (Shannon and Weaver, 1949; Cover and Thomas, 2005). The mutual information $I(M_i; q)$ between a channel (combination) $M_i$ and $q$ is a measure of the information content of $M_i$ with respect to $q$: The higher the mutual information, the more information can be gained from $M_i$ in a retrieval of $q$. We calculate the mutual information $I(M_i; q|C)$ for different sets of conditions $C$ to find the set of conditions $C^*$ which maximizes the mutual information. These optimal sets of conditions are then used for the respective conditional probabilities, $P(M_i|q, C^*)$. A selection of computed mutual information values for different SEVIRI channel (combinations) and sets of conditions is displayed in Fig. 3. To gain insights into the contributions of different channel (combinations) to cloud and phase detection, we additionally calculate the mutual information between each channel $M_i$ and the cloud classification $\mathrm{cloudy/clear}$, as well as between $M_i$ and the phase classification, under the specified conditions $C$. By comparing the mutual information values for $I(M_i; q|C)$, $I(M_i; \mathrm{cloudy/clear}, |, C)$ and $I(M_i; \mathrm{phase}, |, C)$, we can assess the extent to which each channel contributes to the detection of cloudy or clear conditions, as well as to the determination of cloud phase.

In the following, we briefly describe which conditional probabilities are consequently used for the retrieval. We discuss the physical connection between each channel (combination) and the cloud state $q$, and the physical reasons why the chosen conditions for the probabilities might enhance their information content.

### 4.1 Prior

As prior knowledge we use the probability

$$P(q|\mathrm{lat}, \mathrm{lon}, \mathrm{season}). \tag{8}$$

This means that the prior is the probability for each cloud state per latitude, longitude and season, calculated from the five years of collocated data. Besides latitude, longitude and season, the set of ancillary parameters $A$ introduced above in Sect. 4 also

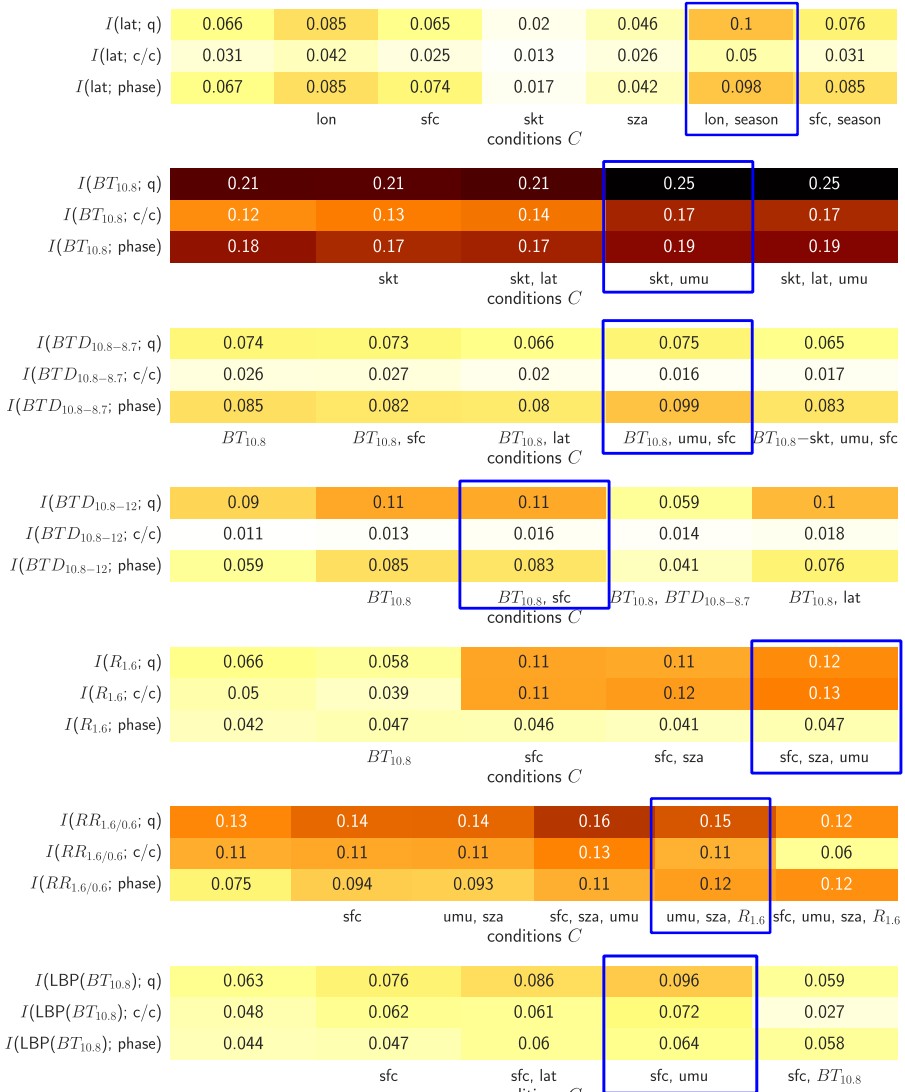

**Figure 3.** First panel: Mutual information $I$ between the latitude and the cloud state $q$ (first row), cloudy/clear, abbreviated as c/c, (second row) and cloud phases (third row) for different sets of conditions $C$. This represents the information content of the different priors we considered, where latitude is a fixed condition, i.e. $P(q\,|\,\mathrm{lat}, C)$. Other panels: Mutual information $I$ between SEVIRI channel (combinations) and cloud state $q$, cloudy/clear and cloud phases for different sets of conditions $C$. Empty spaces for $C$ mean no condition, i.e. the starting point of $I$ before conditions are introduced. The different mutual information values for $q$, cloudy/clear and phase indicate whether a channel (combination) contributes more to cloud or phase detection. The blue boxes indicate the sets of conditions selected for ProPS.

includes surface type, surface temperature, and solar/satellite zenith angles. However, since latitude and longitude are already constrained, incorporating surface type or satellite viewing angles as additional constraints becomes unnecessary. Furthermore,

our mutual information calculations show that conditioning on latitude, longitude and season yields the prior with the optimal
information content compared to other possible sets of conditions (see Fig. 3). This means that location (latitude and longitude) and season are the main dependencies.

## 4.2 Brightness temperature at $10.8\,\mu m$

As the first SEVIRI measurement we use the BT centred at $10.8\,\mu$m wavelength, $BT_{10.8}$, located in the atmospheric window of the electromagnetic spectrum. At this wavelength, the atmosphere is most transparent compared to all other SEVIRI infrared
channels. Therefore, it is a good approximation for the temperature of the surface and (optically thick) cloud tops - one of the most important parameters for cloud detection and phase discrimination. This can also be seen in Fig. 3, as the mutual information between $q$ and $BT_{10.8}$ has the highest values compared to all other SEVIRI channel mutual information values. We use the conditional probability

$$P(BT_{10.8}\,|\,q,\mathrm{umu},\mathrm{skt}). \tag{9}$$

By conditioning on $\mathrm{skt}$ we take into account the temperature difference (contrast) between $BT_{10.8}$ and the surface temperature. This is particularly important for cloud detection. The dependence on $\mathrm{umu}$ is particularly relevant for optically thin clouds, where a higher satellite zenith angle means an effective increase in optical thickness and therefore smaller $BT_{10.8}$ values.

## 4.3 Brightness temperature difference between $10.8\,\mu$m and $8.7\,\mu$m channels

The BTD between the $10.8\,\mu$m and $8.7\,\mu$m window channels is commonly used in phase determination algorithms (Menzel
et al., 2002; Platnick et al., 2003; Zhou et al., 2022). This BTD, denoted as $BTD_{10.8-8.7}$, provides valuable information about cloud phase in several ways. Firstly, it is sensitive to the amount of water vapor present above cloud top. This is because the $8.7\,\mu$m channel is more strongly affected by water vapor absorption in the atmosphere compared to the $10.8\,\mu$m channel. Thus, the BTD is closely related to the cloud top height and thus to the cloud top temperature, which in turn is related to the cloud phase. Secondly, the BTD is influenced by the effective radius of cloud particles (Ackerman et al., 1990). This parameter
provides a clue about the phase of the cloud since ice crystals generally have larger effective radii than liquid droplets. Thirdly, $BTD_{10.8-8.7}$ is sensitive to cloud optical thickness (for small optical thicknesses, Ackerman et al., 1990). On the one hand, this is helpful for the detection of optically thin clouds, on the other hand, this can indirectly indicate the cloud phase since only ice clouds, such as cirrus clouds, typically show very low optical thicknesses. Note, however, that dissipating clouds or a fractional cloud cover can also result in low optical thickness in SEVIRI pixels, which could bias the interpretation of these clouds as ice
clouds. Lastly, the BTD also has a direct dependence on cloud phase for optically thin clouds, i.e. when transmission through the cloud is significant, since the variation in scattering and absorption properties between the two wavelengths $8.7\,\mu$m and $10.8\,\mu$m is different for ice crystals and liquid droplets. We use the conditional probability

$$P(BTD_{10.8-8.7}\,|\,q,BT_{10.8},\mathrm{umu},\mathrm{sfc}). \tag{10}$$

Conditioning on umu takes into account that the satellite zenith angle affects the path length and therefore both the amount of water vapour above the cloud and the effective cloud optical thickness. We also condition on the surface type, since the typical values of $BTD_{10.8-8.7}$ for clear sky differ between surface types - especially for deserts such as the Sahara or the Arabian Peninsula due to the low spectral emissivity of desert dust at $8.7 \mu m$ (Masiello et al., 2014). The relationship with $BT_{10.8}$ is obvious, since it is contained in $BTD_{10.8-8.7}$.

### 4.4 Brightness temperature difference between $10.8 \, \mu$m and $12.0 \, \mu$m channels

The BTD between the two window channels at wavelengths $10.8 \mu$m and $12.0 \mu$m is often used in satellite retrievals for cloud detection and cloud properties (e.g., Key and Intrieri, 2000; Pavolonis et al., 2005; Krebs et al., 2007; Kox et al., 2014; Hünerbein et al., 2022). $BTD_{10.8-12.0}$ is mainly sensitive to optical thickness and effective radius. Both of these quantities contain information about the cloud phase as mentioned above. Furthermore, $BTD_{10.8-12.0}$ also depends directly on the phase, especially for small optical thicknesses, since, as for $BTD_{10.8-8.7}$, the scattering and absorption properties between the two wavelengths $12.0 \mu$m and $10.8 \mu$m vary differently for ice crystals and liquid droplets (Key and Intrieri, 2000). We use the conditional probability

$$P(BTD_{10.8-12.0} \,|\, q, BT_{10.8}, \text{sfc}) \tag{11}$$

Since the main sensitivity is on optical thickness, $BTD_{10.8-12.0}$ is mainly useful for detecting thin ice clouds. This is particularly useful when combined with $BT_{10.8}$, as $BTD_{10.8-12.0}$ can distinguish between warm cloud top temperatures and optically thin clouds with warm surface temperatures, which may have the same value of $BT_{10.8}$.

### 4.5 Reflectivity of the $1.6 \, \mu m$ channel

The reflectivity of solar radiation is generally a good indicator for the presence of a cloud, as clouds are usually brighter (more reflective) than the surface for clear sky conditions. Further, near-infrared (NIR) reflectivity, like the $1.6 \mu$m channel, is a well established indicator of cloud phase as the reflectivity at $1.6 \mu$m, $R_{1.6}$, is sensitive to the effective radius of cloud particles: The typically small liquid droplets reflect more radiation at this wavelength than the typically large ice crystals. In addition to its sensitivity to the effective radius, $R_{1.6}$ is also sensitive to the phase itself, since ice absorbs more radiation than water at this wavelength. We use the conditional probability

$$P(R_{1.6} \,|\, q, \text{sza}, \text{umu}, \text{sfc}). \tag{12}$$

Conditioning on the solar and satellite zenith angles, sza and umu, takes into account that reflectivities are angle dependent. The sensitivity of $R_{1.6}$ on azimuth angles is comparatively small, we therefore neglect it in order to keep the number of conditions small. The surface type, sfc, is a proxy for surface albedo, as different surface types each have typical albedo values.

## 4.6 Reflectivity ratio of the $0.6\,\mu m$ and $1.6\,\mu m$ channels

As the next observation, we consider the reflectivity ratio $RR_{1.6/0.6} = \frac{R_{1.6}}{R_{0.6}}$. The combination of a NIR channel ($R_{1.6}$) and a visible channel ($R_{0.6}$) is often used to retrieve cloud microphysical parameters such as effective radius and optical thickness (Nakajima and King, 1990). These microphysical parameters contain phase information, so combining NIR and visible channels is useful for a phase retrieval (Knap et al., 2002; Marchant et al., 2016). We use the ratio between the two channels to reduce the dependence on solar and satellite viewing angles as well as on particle number concentration (Chylek et al., 2006). We use the probability

$$P(RR_{1.6/0.6} \,|\, q, R_{1.6}, \mathrm{sza}, \mathrm{umu}). \tag{13}$$

Apart from the dependence on $R_{1.6}$, we again consider the solar and satellite zenith angles for the same reasons as for the conditional probabilitiy of $R_{1.6}$.

## 4.7 Local binary pattern at $10.8\,\mu m$

Finally, we use the local binary pattern (LBP) of the $10.8\,\mu m$ infrared channel, $\mathrm{LBP}(BT_{10.8})$. The LBP technique is used for texture analysis. It characterizes the spatial variations of pixel intensities by comparing the central pixel with its surrounding neighbors within a defined local region. Texture parameters have already been used in Bayesian retrieval methods for cloud detection (Merchant et al., 2005). The texture of clouds differs in most cases from the texture of the surface, so that the LBP can help in the detection of clouds. Further, the texture of cloudy regions can differ for different cloud types, such as small cumulus clouds with large local variations in reflectivity versus large smooth cirrus clouds with small variations in reflectivity. Since different cloud types are associated with different cloud phases, the LBP is also a suitable parameter for phase detection.

To compute the LBP, the central pixel is compared with eight surrounding pixels in a defined neighbourhood: if the intensity value of a neighbour is greater than or equal to the intensity of the central pixel, a binary 1 is assigned; otherwise, a binary 0 is assigned for each neighbour. The sum of these binary values contains valuable texture information: the maximum sum value of 8 indicates a uniform image region, while lower values indicate non-uniform regions. For example, a sum of 4 indicates an even distribution of neighbours with both higher (or equal) and lower intensities compared to the central pixel. A Gaussian filter is then applied to smooth the results to obtain a continuous value.

The infrared channel $BT_{10.8}$ is well suited for calculating a texture as the atmosphere is most transparent at this wavelength compared to all other SEVIRI infrared channels. The advantage of choosing an infrared channel is that it is also available during the night. The LBP of $BT_{10.8}$ is particularly useful for detecting low clouds during the night, which are otherwise difficult to distinguish from clear sky for infrared channels. We use the conditional probability

$$P(\mathrm{LBP}(BT_{10.8}) \,|\, q, \mathrm{sfc}, \mathrm{umu}). \tag{14}$$

The condition on surface type, sfc, takes into account that different surface types have different textures. The condition on umu takes into account that pixel sizes and therefore the computed texture from LBP vary with umu.

## 5  The PRObabilistic cloud top Phase retrieval for Seviri (ProPS)

This section gives an overview of the ProPS retrieval method using the equations and probabilities explained in the last sections (Sect. 3 and Sect. 4). Figure 2 gives a schematic overview of the retrieval method.

### 5.1  Cloud top phase

The output of the Bayesian method is a probability $P(q\,|\,M, A)$ for each cloud state $q \in \{\text{clear}, \text{TI}, \text{IC}, \text{MP}, \text{SC}, \text{LQ}\}$, of which we use the cloud state with the highest probability, $q^*$, as the final result.

### 5.2  Daytime

Using the probabilities for the selection of SEVIRI channels as explained in the previous section, the cloud state retrieval equation for ProPS (see Eq. 3) becomes

$$P(q\,|\,M, A) = \frac{1}{N} P(\text{LBP}(BT_{10.8})\,|\,q, \text{sfc}, \text{umu})$$

$$P(RR_{1.6/0.6}\,|\,q, R_{1.6}, \text{sza}, \text{umu})$$

$$P(R_{1.6}\,|\,q, \text{sza}, \text{umu}, \text{sfc})$$

$$P(BTD_{10.8-12.0}\,|\,q, BT_{10.8}, \text{sfc})$$

$$P(BTD_{10.8-8.7}\,|\,q, BT_{10.8}, \text{umu}, \text{sfc})$$

$$P(BT_{10.8}\,|\,q, \text{umu}, \text{skt})\,P(q\,|\,\text{lat}, \text{lon}, \text{season}) \tag{15}$$

with the normalization factor $N = N(M, A)$ defined such that $\sum_q P(q\,|\,M, A) = 1$. $M$ is the set of SEVIRI channel (combinations)

$$M = \{\text{LBP}(BT_{10.8}), RR_{1.6/0.6}, R_{1.6}, BTD_{10.8-12.0},$$

$$BTD_{10.8-8.7}, BT_{10.8}\} \tag{16}$$

and $A$ the set of ancillary parameters (see Eq. 7).

### 5.3  Nighttime

During the night, only thermal SEVIRI channels are available. For the night version of ProPS we therefore only use probabilities of the thermal channels from Eq. 15:

$$P(q\,|\,M, A) = \frac{1}{N} P(\text{LBP}(BT_{10.8})\,|\,q, \text{sfc}, \text{umu})$$

$$P(BTD_{10.8-12.0}\,|\,q, BT_{10.8}, \text{sfc})$$

$$P(BTD_{10.8-8.7}\,|\,q, BT_{10.8}, \text{umu}, \text{sfc})$$

$$P(BT_{10.8}\,|\,q, \text{umu}, \text{skt})\,P(q\,|\,\text{lat}, \text{lon}, \text{season}) \tag{17}$$

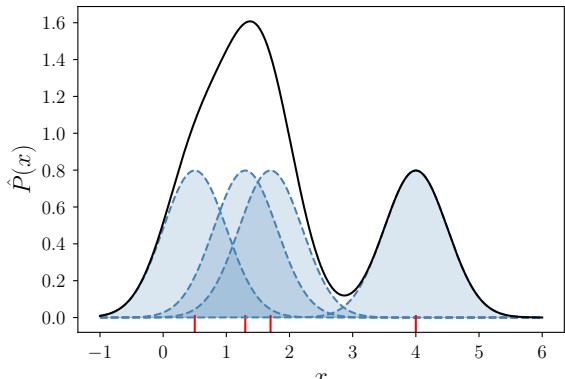

**Figure 4.** Construction of a kernel density estimate (continuous line) with a Gaussian kernel (dashed lines) for four samples of the true probability distribution (vertical red line segments). Figure adapted from Węglarczyk (2018) (CC BY 4.0 https://creativecommons.org/licenses/by/4.0/).

## 6  Computation of probabilities

We use the method of kernel density estimation (KDE) to compute the probabilities needed for ProPS from the collocated data set. KDE is a technique for estimating a probability density function (pdf), which better represents the details of the pdf compared to traditional histograms (Węglarczyk, 2018). The KDE technique provides a smooth estimate of the pdf without imposing assumptions about its shape. Further advantages are that, unlike histograms, it includes all sample point locations and can more convincingly suggest the presence of multiple modes (Węglarczyk, 2018). Consider a variable of interest $x$ with

an unknown probability distribution $P(x)$ and a sample of $n$ observations, $x_1, x_2, ...x_n$, of that variable. To compute the kernel estimate $\hat{P}(x)$ of the true probability distribution $P(x)$, we assign a kernel function $K(x_i, x)$ to each sample data point $x_i$ as follows (Silverman, 1986; Węglarczyk, 2018):

$$\hat{P}(x) = \frac{1}{n} \sum_{i=1}^{n} K(x_i, x). \tag{18}$$

The kernel function $K(x_i, x)$ is centred at $x_i$ and normalised to unity, i.e. $\int_{-\infty}^{+\infty} K(x_i, x)dx = 1$. We employ a Gaussian kernel

function, which is commonly used. The kernel transforms the discrete point location represented by $x_i$ into a smooth distribution centred around $x_i$. Figure 4 illustrates this technique for the one dimensional case. For $d > 1$ dimensions, both $x$ and $x_i$ become $d$-dimensional vectors instead of scalars. For example, in our case, to compute the probability $P(BT_{10.8}, q, \mathrm{umu}, \mathrm{skt})$, the variable $x$ is a four-dimensional vector $x = (BT_{10.8}, q, \mathrm{umu}, \mathrm{skt})$.

The width of the kernel function determines the amount of smoothing and is represented by a parameter called bandwidth

$h$. Too small values of $h$ may result in a probability estimate showing insignificant details, while too large values of $h$ may smooth out important features (Węglarczyk, 2018). A certain compromise is needed. We choose to use an (effectively) dynamic bandwidth $h$, since there are regions of parameter space with many samples that allow small values of $h$, and other regions

with few samples that require large $h$: before computing the kernel estimate $\hat{P}(x)$, the variable $x$ is transformed, $x^t = f(x) :=$ $\arctan(\frac{1}{\beta}(x-\alpha))/\gamma$. As a non-linear transformation, $f(x)$ can reshape the distribution of the data by stretching or compressing

certain regions by fine-tuning the $\alpha$, $\beta$ and $\gamma$ parameters. The parameters of the transformation are chosen for each variable $x$ in such a way that the samples of the variable $x_i$ are more evenly distributed in the transformed space. The arctan function in the transformation is particularly useful for this purpose, as it has the ability to condense the edges of parameter space, where there are typically fewer samples, while expanding the central region. The parameters $\alpha$ and $\beta$ can be understood as the global mean and variance of the variable $x$. Additionally, these transformation parameters are chosen to ensure that all transformed

variables fall within a similar range, typically around $-1$ to $1$, to maintain similar smoothness in the directions of all variables. This requires (in some cases) linear scaling with the $\gamma$ parameter in the transformation function. After the transformation, the kernel estimate $\hat{P}^t(x^t)$ is computed in the transformed space using a constant bandwidth. The variable is finally transformed back to the original variable space, $\hat{P}^t(x^t) = \hat{P}^t(f(x)) =: \hat{P}(x)$. This approach results in a narrower kernel in regions with many $x_i$ samples and a wider kernel in regions with fewer $x_i$ samples. Consequently, our procedure allows for detailed features

in the kernel estimate $\hat{P}(x)$ where numerous samples are available, while maintaining reasonable smoothness and flatness in regions with limited samples. The transformation parameters as well as the bandwidth for each variable are shown in table 1.

In the case of discrete variables such as $q$, season, or surface type, the KDE method cannot be used directly. Instead, we divide the variable space into subcategories based on all possible combinations of the discrete variables of the probability in question. For each subset, we utilize the KDE method to calculate the probability for the continuous variables within

that specific subcategory. Subsequently, we normalize the probabilities to obtain a normalized probability distribution that incorporates both discrete and continuous variables.

From the so computed kernel estimate $P(x)$ with $x$ a $d$-dimensional vector $x = (X^1, X^2, ...X^d)$ a conditional probability can be computed using the relationship

$$
\begin{aligned}
P(X^1 | X^2, ..., X^d) &= \frac{P(X^1, X^2, ..., X^d)}{P(X^2, ..., X^d)} \\
&= \frac{P(X^1, X^2, ..., X^d)}{\sum_{X^1} P(X^1, X^2, ..., X^d)}.
\end{aligned}
\tag{19}
$$


The probabilities are only computed for the locations in parameter space where a sufficient number of samples, $x_i$, are available. If too few samples are available, the pdf is set to a flat distribution, i.e. it contains no information and does not change the probability for the cloud state $q$ when multiplied as in the retrieval Eq. 15. Since the collocated data set is quite large, this is only necessary for a few special cases. Most notably, this is necessary for the solar channel (combination) $R_{0.6}$

and $RR_{1.6/0.6}$ for the regions of sza-umu parameter space where no samples are available (see Sect. 2.2 and Fig. 1). There is however one important special case for the probabilities of the solar channel (combination) $R_{0.6}$ and $RR_{1.6/0.6}$, for which we proceed differently: DARDAR data are not available for sza values below $20°$ (see Sect. 2.2), as the sun-synchronous orbit of the polar orbiting satellites Calipso/CloudSAT never reaches lower sza values. For these relatively low sza values, the dependence of the reflectivity on sza is small compared to other dependencies. As a simple solution for this special case, we therefore use

the probabilities calculated for the lowest available sza also for the smaller values of sza.

**Table 1.** Parameters for transforming and computing the kernel density estimate (KDE) for SEVIRI measurements and ancillary parameters

| variable | transformation parameters | bandwidth |
|---|---|---|
| $BT_{10.8}$ | $\alpha = 270, \beta = 30, \gamma = 1$ | 0.04 |
| $BTD_{10.8-8.7}$ | $\alpha = 2.3, \beta = 2, \gamma = 1.5$ | 0.04 |
| $BTD_{10.8-12}$ | $\alpha = 1, \beta = 3, \gamma = 1.1$ | 0.04 |
| $R_{1.6}$ | $\alpha = 30, \beta = 40, \gamma = 1$ | 0.04 |
| $RR_{1.6/0.6}$ | $\alpha = 0.7, \beta = 1.1, \gamma = 1$ | 0.04 |
| LBP($BT_{10.8}$) | $\alpha = 6, \beta = 2, \gamma = 1$ | 0.04 |
| sza | $\alpha = 45, \beta = 120, \gamma = 1$ | 0.04 |
| umu | $\alpha = 0.58, \beta = 1.2, \gamma = 1$ | 0.04 |
| skt | $\alpha = 290, \beta = 20, \gamma = 1$ | 0.04 |
| lat | no transformation | 2 |
| lon | no transformation | 2 |

Using this KDE method, we compute all probability distributions needed for the ProPS algorithm (see Eq. 15). Fig. 5 shows an example for the probability $P(BT_{10.8} \,|\, q, \mathrm{umu}, \mathrm{skt})$, i.e. the probability to measure $BT_{10.8}$ values, given the cloud state $q$ (in different colors) and fixed values for the surface temperature, skt, and satellite zenith angle, umu. As expected, for clear sky the probability peaks at $BT_{10.8}$ values close to the surface temperature. For LQ, SC, MP and IC clouds, the probability distribution shifts to increasingly lower $BT_{10.8}$ values. There are however large overlap regions, which show that the cloud state cannot be determined from $BT_{10.8}$ measurements alone. TI clouds have a relatively flat probability distribution over a wide range of $BT_{10.8}$ values, since the radiation from the surface is transmitted to a varying degree. More examples for probability distributions can be found in the appendix (see Fig. A1).

## 7 Example application of ProPS

Figure 6 (right) shows the output of the ProPS retrieval for one exemplary SEVIRI scene, on 2022-04-25 at 12:00 UTC. For comparison the natural color RGB of the scene is also shown (left). The result of the ProPS retrieval looks sensible: The retrieval detects (most) clouds which can be seen in the RGB. The distribution of phases on the SEVIRI disc makes physical sense, with e.g. mainly IC in the Intertropical Convergence Zone (ITCZ), LQ over the subtropical ocean and SC/MP mainly over the Southern Ocean and Northern high latitudes.

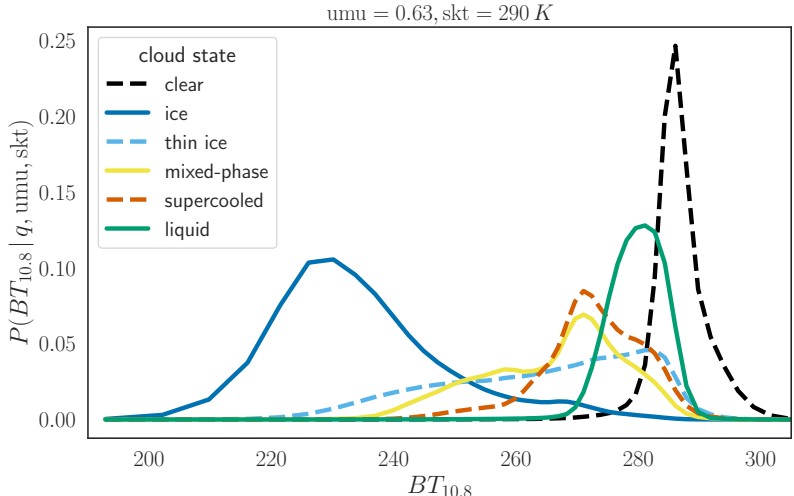

**Figure 5.** Example for the probability distribution $P(BT_{10.8} \,|\, q, \mathrm{umu}, \mathrm{skt})$ computed using KDE at fixed values for umu and skt.

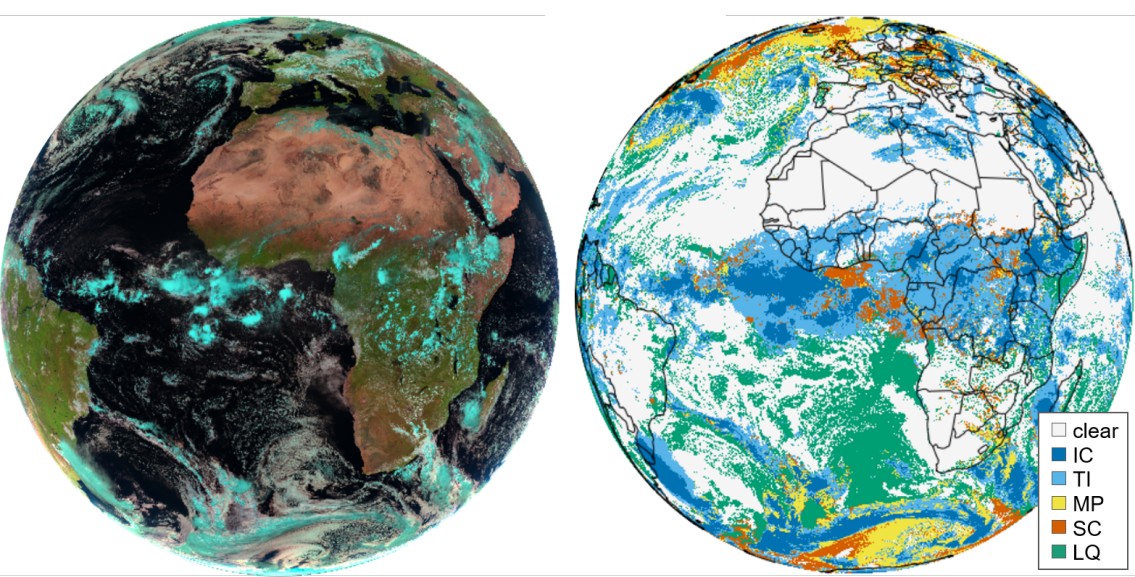

**Figure 6.** RGB composite (left) and example application of ProPS (right) for a SEVIRI scene on the 2022-04-25 at 12:00 UTC.

 ## 8  Performance evaluation using DARDAR

In this section we evaluate how well ProPS is able to reproduce the DARDAR cloud detection and phase classification. To this end we randomly select six months from the five years collocated data set as validation data set (constraining that every season

is represented), which amounts to about 3.7 million data points. These data points of the validation data set are not used for the computation of the probabilities (see Sect. 6) to perform an independent validation.

## 8.1 Comparison to DARDAR example tracks

We start the performance evaluation with two example curtains from DARDAR to highlight the strengths of the ProPS retrieval and the challenges posed by, for example, complex cloud scenes or the different viewing geometries of polar orbiting and geostationary satellites (see Fig. 7). These two examples demonstrate how the retrieval works at different latitudes and meteorological conditions. Both figures show a DARDAR curtain coarsened to SEVIRI resolution and the corresponding results of the ProPS algorithm in the plots above, i.e. probabilities for the cloud state $q$ and the certainty measure along the track. As an overlay on the DARDAR curtain, the figures show the most likely cloud state from ProPS, $q^*$, and the cloud state retrieved from DARDAR, $q_{\mathrm{dardar}}$, which is an aggregate of all DARDAR values per SEVIRI pixel over a vertical depth of 240 m from the cloud top (see Sect. 2.1 and Mayer et al. (2023) for details).

The ProPS and DARDAR cloud states, $q^*$ and $q_{\mathrm{dardar}}$, match well in most cases. For the high latitude example in Fig. 7(a), ProPS is able to detect MP and SC clouds even for very low (< 1 km) cloud top heights. Figure 7(b) shows that MP and SC clouds are also present in low latitudes close to the equator where convection is the main cloud formation mechanism and that ProPS is mostly able to detect them. This might be very useful for future studies of the life cycle and phase transitions of convective clouds (Coopman et al., 2020). The two figures also show some examples of small cirrus clouds as well as some LQ clouds beneath an aerosol layer. In both cloud situations, clouds are mostly retrieved in an accurate way. In general however, the detection works best for spatially extended cloud states. The probabilities for the cloud state, $P(q)$, and the corresponding certainty measure show that some clouds can be classified "more easily" than others, i.e. when the probability for a particular state is close to one, corresponding to high values of the certainty parameter. This is the case, for example, for the large IC clouds and some LQ clouds and clear sky pixels in the example figures.

However, the examples highlight also challenging situations for the retrieval: In the DARDAR curtain SC and MP cloud tops often appear together in a cloud and alternate on small spatial scales. ProPS is often not able to resolve this small scale variability. Another challenge is posed by optically thin ice clouds. When ProPS fails to detect these TI clouds, it often classifies these pixels either as the cloud state of a cloud below, if the overlying TI cloud is optically very thin so that the radiation from the cloud below is largely transmitted through the overlying ice cloud, or as MP, if the overlying TI cloud is somewhat thicker and the radiation signals from a cloud below containing liquid particles mix with the overlying TI cloud signal. This effect often happens at the edges of large ice clouds, which are typically optically very thin and/or do not fill an entire SEVIRI pixel. An example can be seen in Fig. 7(a) at the edges of the large ice cloud on the right. To overcome this shortcoming, a combination of ProPS with a cloud product that identifies multilayered clouds would make sense in the future (as is for instance planned for the EarthCARE multi-spectral imager Hünerbein et al., 2022). Another challenge, again related to optically thin clouds, is the misclassification of MP, SC or LQ clouds as TI when they are optically thin, e.g. during formation or dissipation. These optically thin clouds are typically characterised by high values of $BTD_{10.8-12}$. Since the vast majority of pixels with high

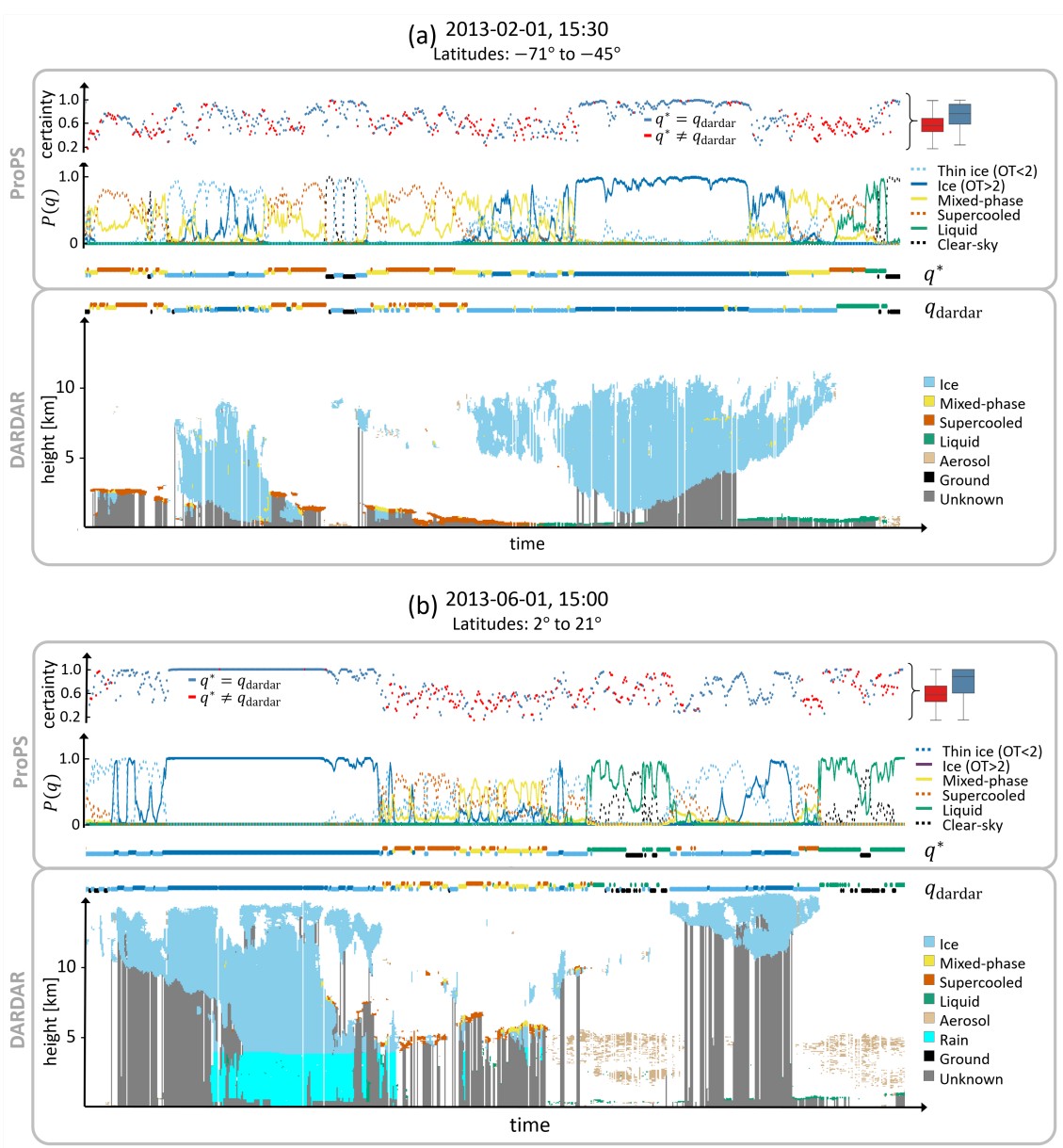

**Figure 7.** Example application of ProPS to DARDAR tracks in (a) high latitudes and (b) low latitudes. The bottom panel of each sub-figure shows the DARDAR curtain coarsened to SEVIRI resolution; the corresponding results of the ProPS algorithm (probabilities $P(q)$) are shown in the panels above. The cloud state retrieved from DARDAR, $q_{\mathrm{dardar}}$, and the most likely cloud state from ProPS, $q^*$, along the track are shown in between (in the same colour code as $P(q)$). Above the $P(q)$ panels, the corresponding certainty of the ProPS results are shown, with the color code indicating whether $q^*$ agrees with $q_{\mathrm{dardar}}$. The box plots on the right show the quartiles of the certainty measure for disagreement ($q^* \neq q_{\mathrm{dardar}}$; red) and agreement ($q^* = q_{\mathrm{dardar}}$; blue).

$BTD_{10.8-12}$ values correspond to TI clouds, ProPS, being a statistical method, tends to label pixels with high $BTD_{10.8-12}$ values as TI clouds.

Sometimes, the ProPS $q^*$ is spatially slightly shifted against the DARDAR results, especially in the high latitude example in Fig. 7(a) where $q^*$ is in some cases slightly shifted to the left relative to $q_{\mathrm{dardar}}$. This is most likely due to the different viewing geometries of the two instruments. Further, as SEVIRI looks at the clouds under a given angle, a high cloud can cover a neighbouring lower cloud from SEVIRI's perspective. In addition, the cloud cover in the rest of the SEVIRI 2D pixel can be different from that in the overflight swath of the polar orbiting satellite, and there can be a time difference of up to 7.5 minutes between the satellites. These effects could explain some of the differences between the ProPS and DARDAR classifications, especially for high certainty pixels where we expect the classification to be correct. However, these effects are difficult to account for in a quantitative evaluation (see Sect. 8.2) and lead to lower probabilities of detection.

The example figures also demonstrate that the cloud situation is often complex, with multi-layered clouds at different altitudes, cloud phase changes on small scales, and other atmospheric factors such as aerosols. The certainty parameter can be an indicator of the complexity of the scene: Complicated cloud scenes, such as multi-layered clouds or rapidly changing phases on small scales, tend to have lower certainty values compared to simpler scenarios. For example, the certainty drops from almost one to lower values in Fig. 7(a) to the left and right of the thick ice cloud, where it becomes thinner with underlying liquid layers.

To get an impression how ProPS compares to other cloud and phase retrieval algorithms, we additionally conducted a comparison of ProPS with the most recent version of the CM SAF CLoud property dAtAset using SEVIRI - Edition 3 (CLAAS-3) for 12 exemplary scenes. CLAAS-3 distinguishes between clear sky, warm liquid, supercooled liquid and ice clouds. We find a good general agreement between the two methods with differences mainly constrained to cloud edges and the transition regions between different phases. In general, ProPS classifies more pixels as cloudy than CLAAS-3, especially small, warm cumulus clouds and categorizes more pixels as thin ice than CLAAS-3. A detailed discussion can be found in the appendix (see Fig. B1 and Fig. B2).

## 8.2 POD and FAR

In the following, we only consider pixels with a homogeneous cloud state over at least three consecutive pixels along the DARDAR curtain. It is difficult for SEVIRI to resolve the cloud state on smaller scales, as mentioned in the section above. Furthermore, isolated cloud state pixels may be artefacts of the DARDAR product, which we try to exclude.

Fig. 8 shows the overall performance of ProPS evaluated pixel by pixel against the DARDAR cloud state for the six months of validation data. We distinguish between cloud and phase detection. Fig. 8(a,c) show the number of clear and cloudy pixels according to DARDAR and colour coded how many of these pixels are identified as clear or cloudy by ProPS. The upper row shows this validation for the daytime version, the lower row for the nighttime version of ProPS. The probability of detection (POD) of clouds (clear sky) is defined as the percentage of pixels classified as cloudy (clear) by both ProPS and DARDAR relative to the pixels classified as cloudy (clear) by DARDAR. With this definition, the POD for clear sky is 86%, for clouds it is 93%. Optically thin TI clouds and small warm LQ clouds are the clouds which are most difficult to detect: of all undetected

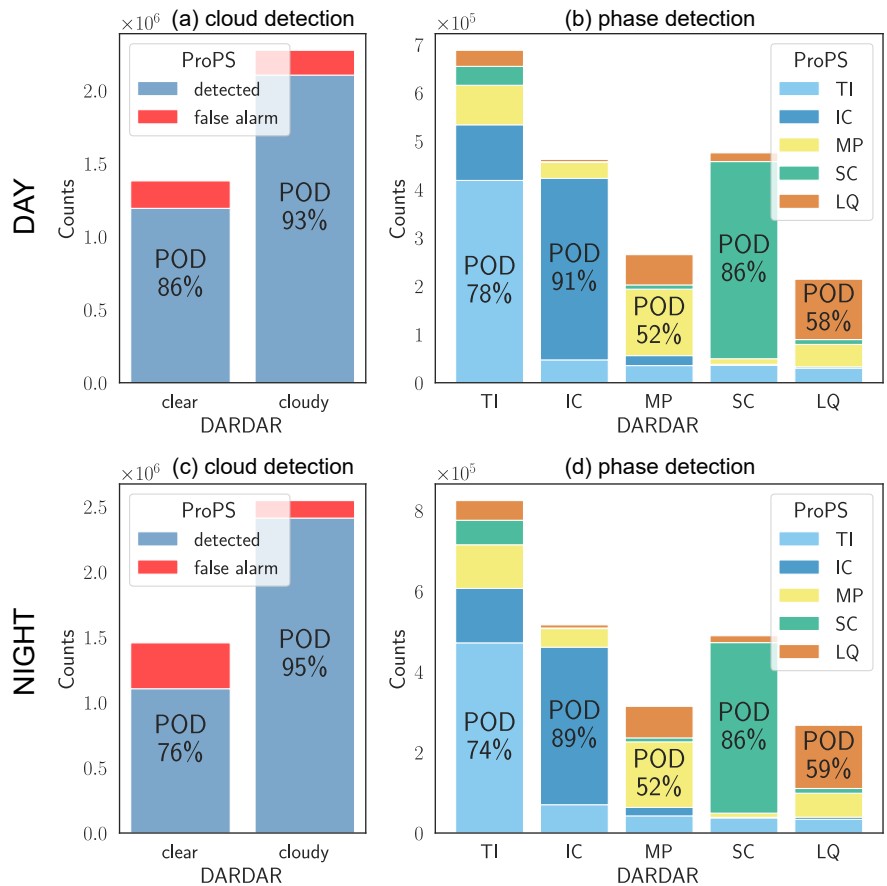

**Figure 8.** Cloud and phase detection for the day version (upper row) and the night version (lower row) of the ProPS method. For IC and TI, we count both ice classifications as correct in the POD values.

clouds (i.e. the red part of the "DARDAR Cloudy" bar in Fig. 8(a)), 54% are TI clouds and 37% are LQ clouds. Difficulties to detect TI clouds are expected since passive sensors are less sensitive to optically thin clouds than Lidar instruments. LQ clouds are particularly difficult to detect when they occur over bright surfaces or are embedded in (thick) aerosol layers. Small LQ clouds that do not fully cover SEVIRI pixels and therefore go undetected also play a role. For the same reasons, TI and LQ are again the two most problematic cloud phases when looking at false alarms: Of all false alarms (i.e. the red part of the "DARDAR Clear" bar in Fig. 8(a)), 40% are classified by ProPS as TI and 43% as LQ clouds. Looking at these results the other way around, this also implies that one can be very sure that there really is a cloud at pixels classified as SC, MP or IC by ProPS during the day and that pixels classified as clear by ProPS are almost never SC, MP or IC clouds.

As expected, the nighttime version of ProPS performs slightly worse than the daytime version, with a POD of 76% for clear sky and 95% for clouds. The nighttime version tends to classify too many pixels as cloudy (red part of the "DARDAR Clear"

bar in Fig. 8(c)). This is particularly the case for LQ clouds, which have similar temperatures as the surface and are therefore difficult to detect using thermal channels alone.

Fig. 8(b) and (d) show the phase detection performance of ProPS for the pixels that are correctly classified as cloudy by the daytime and nighttime version of ProPS respectively. The POD is defined analogously as for cloud detection. For the daytime version, the POD for IC, TI, MP, SC and LQ is 91%, 78%, 52%, 58% and 86% respectively. For the calculation of these POD values, for IC (TI) clouds, the other ice classification, TI (IC), was also counted as correctly classified, since it is the same thermodynamic phase. The POD values show that the majority of pixels is correctly classified by ProPS. The phase classification works especially well for IC and LQ clouds. The TI clouds which are not correctly classified by ProPS are mainly optically very thin TI clouds with other clouds below. As explained in Sect. 8.1 these pixels are often classified either as MP or as the cloud phase of the cloud below. Fig. 8(b) shows that it is difficult to distinguish between MP and SC, with many MPs being classified as SC and vice versa. This difficulty is expected since SC and MP cloud tops occur in very similar circumstances (similar latitudes, cloud top temperatures and cloud types) and alternate on relatively small scales (see Fig. 7). In addition, an MP cloud top may consist mainly of liquid droplets and therefore have very similar radiative properties to an SC cloud top. Unfortunately, there is no parameter quantifying the liquid fraction of MP pixels in DARDAR, so we have no way of checking the performance of ProPS MP detection as a function of liquid fraction. Nevertheless, results show the ability of ProPS to identify also the most challenging phases MP and SC (more than half of the DARDAR MP and SC pixels are correctly classified by ProPS, see numbers above).

Interestingly, the nighttime phase classification performs remarkably well, almost on par with the daytime version. To understand why this is the case we studied examples in the SEVIRI disc and compared the phase classification using only thermal channels against using only solar channels for the retrieval. We find that there are "easier", unambiguous cloud phase cases, for which the classification using only thermal or only solar is in both cases correct and hence in these situations the combination of thermal and solar channels does not lead to different results. For the more complex cases, the classification is challenging for both thermal and solar channels and the combination of solar and thermal information does not lead to a significant increase of correctly detected phases. However, the certainty of the retrieval increases considerably when all channels are used. Since solar channels contain valuable information on the phase, as outlined in Sect. 4, the increase in certainty when using all channels shows that the solar channels indeed enhance the accuracy of phase determination while boosting the confidence of the obtained results. It has also been shown in previous studies that the use of solar channels increases accuracy in phase detection (Baum et al., 2000). Note that the similar performance of the two algorithm versions is only true if we consider the cases where a cloud has been correctly (according to DARDAR) detected. For cloud detection, thermal and solar channels have complementary advantages: Solar channels are very helpful at detecting low clouds, which have similar temperatures as the surface, while thermal channels have advantages for detecting optically very thin clouds. Therefore, the combination of the selected thermal and solar channels is the best option for a reliable cloud and phase detection, but the similarity of the performance of ProPS during daytime and nighttime allows for a smooth transition from day to night.

Recall that the output of ProPS contains not only the most likely cloud state, $q^*$, but also the probabilities for all cloud states. In cases where $q^*$ does not match DARDAR, the second most likely cloud state often does. This is especially true for MP and

510 SC clouds: When $q^*$ does not match the DARDAR classification of MP (SC), 68% (65%) of these pixels have MP (SC) as their second most likely cloud phase. Hence, if both the most and second most likely cloud states are considered correct, the POD increases to 84% for both MP and SC. This means that we can gain information from the second most likely cloud state result.

## 8.3 Relation to the certainty parameter

One of the advantages of the Bayesian approach is the certainty parameter for the retrieval (see Sect. 3.3). For the example

curtains in Fig. 7, the mean certainty values are shown on the right for pixels where ProPS and DARDAR agree or disagree. Where ProPS and DARDAR agree, the average certainty is higher, indicating that the certainty measure is meaningful. However, as the examples in Fig. 7 show, this is only true on average - there are still cases with a low level of certainty that are correctly identified, and vice versa.

Figure 9 gives an overview of the relation to the certainty parameter for the six months of validation data for the ProPS day

version. It shows the POD and false alarm rate (FAR) for cloud detection and phase determination (given a detected cloud) for each phase separately and their average (weighted by the counts of each phase) per certainty bin of width 0.1. The two lower panels show the number of occurrences of the certainty values. The average POD for cloud detection is high ($> 90\,\%$) for almost all certainty values; the FAR decreases monotonically with increasing certainty. This means that ProPS tends to overestimate cloud amount at low certainty values, as also mentioned in Sect. 8.2, but has an increased detection accuracy at

higher certainty values. For phase determination, the average POD increases monotonically with the certainty parameter, while the average FAR decreases. Hence, the certainty parameter is a useful tool to decide whether to trust a result.

From the number of occurrences of certainty values (lower panels in Fig. 9) and examples as in Fig. 7 we see that the most unambiguous cases are clear sky, IC and LQ clouds (if their spatial extent is large enough to fill whole SEVIRI pixels). MP, SC and TI clouds have on average lower certainty values than the other cloud states.

## 530 8.4 Performance on the SEVIRI disc

To better characterize the performance of ProPS, we evaluate its POD on the SEVIRI disc for the six months of validation data. This evaluation is shown in Fig. 10 and Fig. 11 for the cloud detection and phase detection given a detected cloud respectively. Here we show the results for the daytime version; the results for the nighttime version can be found in the appendix (see Fig. C1 and Fig. C2). The top panels show the POD of each cloud state and the lower panels show the corresponding distribution of

535 the number of occurrences of each cloud state according to DARDAR.

Fig. 10 shows that cloud detection is most challenging over deserts, such as in northern and southern Africa. Clear sky detection is most challenging at the ITCZ and some regions in high latitudes. Looking at the distribution of occurrences, it can be seen that the regions where cloud and clear sky detection are most challenging correspond to the regions with the fewest occurrences of each.

The same is mostly true for phase detection of TI, MP, SC and LQ (see Fig. 11). For instance, MP and SC have their highest detection rates in high latitudes where they occur most often. The detection of IC clouds on the other hand is uniformly high over the whole SEVIRI disc.

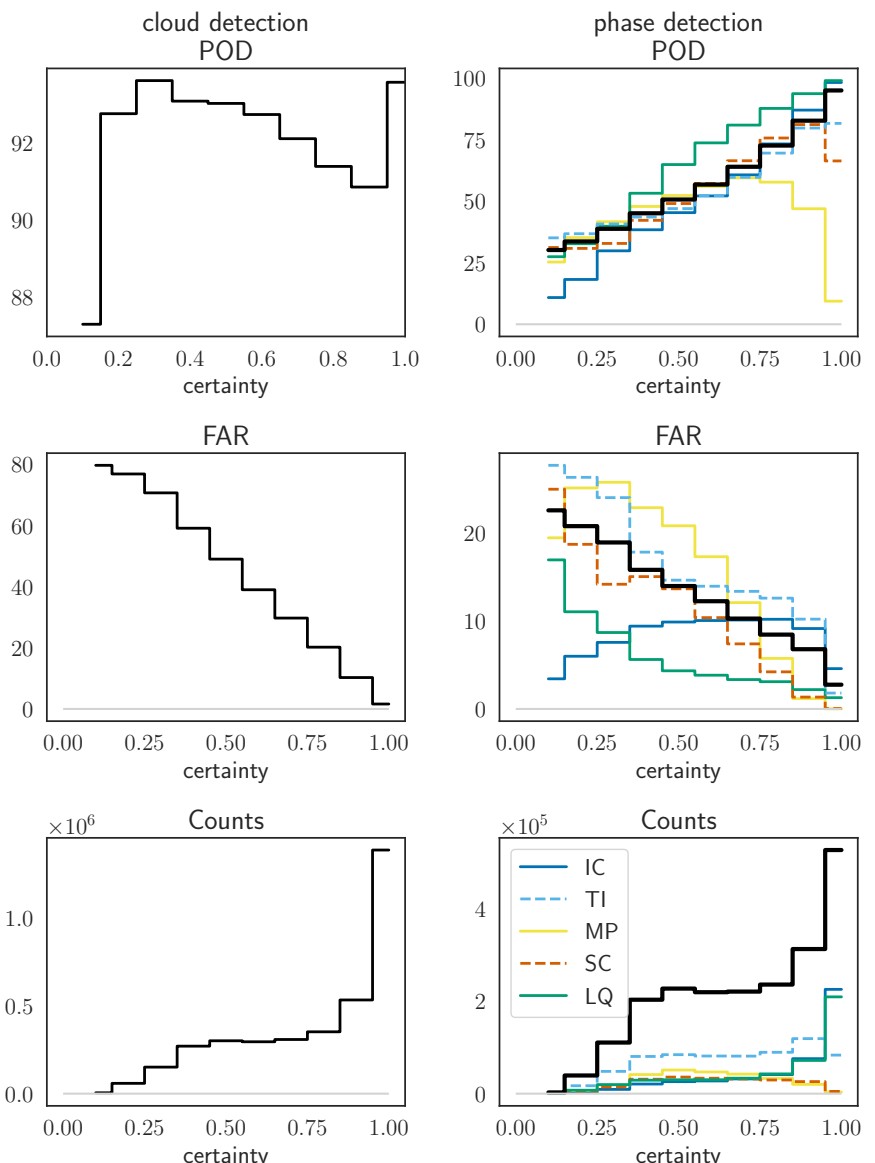

**Figure 9.** Top row: POD of cloud and phase detection (given that a cloud was detected) for each phase separately (in colour) and their weighted average (in black) as a function of the certainty parameter. Middle row: FAR for cloud and phase detection. Lower row left: Number of occurrences of certainty values. Lower row right: Number of occurrences of certainty values given a cloud was detected (in black) and the contributions from each phase (as classified by DARDAR; in colour).

For the night version of ProPS, the POD of clouds is similar to the day version while the POD of clear sky is slightly lower almost everywhere in the SEVIRI disc (see Fig. C1). This suggests that ProPS tends to overestimate cloudiness during the
545 night. The spatial distribution of the POD of the different phases is very similar to the daytime version (see Fig. C2).

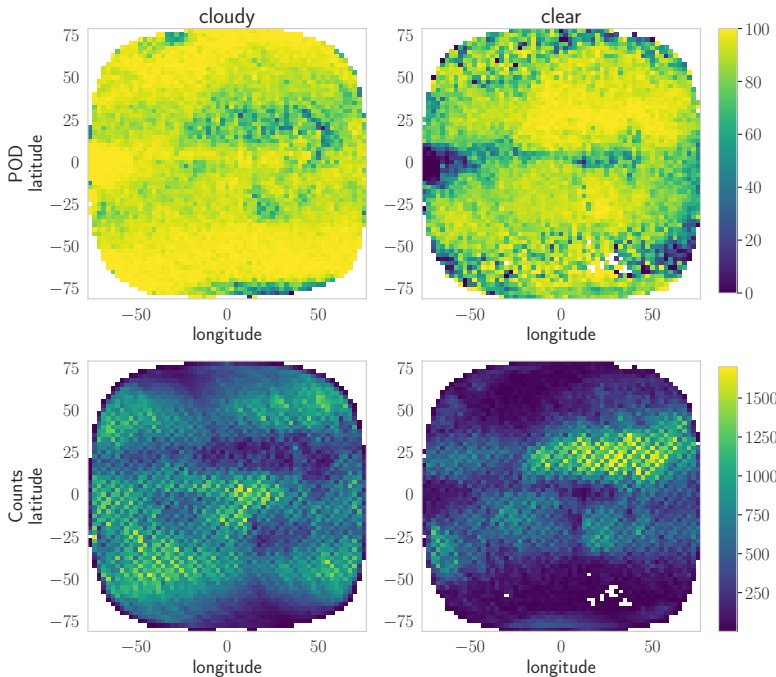

**Figure 10.** POD (upper row) and counts of occurrences (lower row) of cloudy and clear sky pixels in the SEVIRI disc for the ProPS day version. POD and counts are computed in latitude-longitude bins of $2.5° \times 2.5°$ for the six months of validation data.

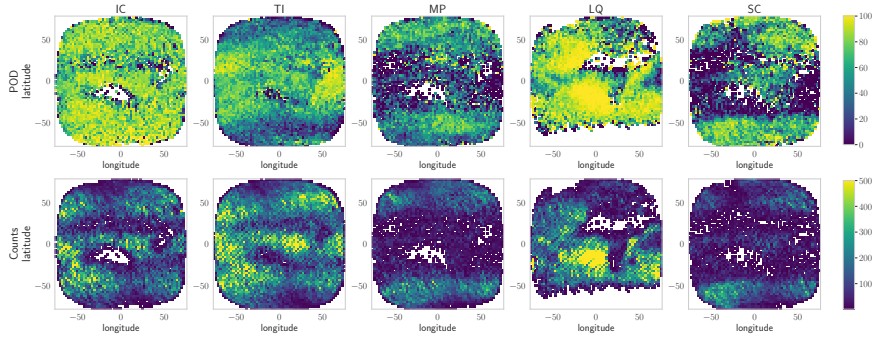

**Figure 11.** POD (upper row) and counts of occurrences (lower row) of the different phases in the SEVIRI disc for the ProPS day version. POD and counts are computed in latitude-longitude bins of $2.5° \times 2.5°$ for the six months of validation data.

## 9 Conclusions

This study presents ProPS, a new method for cloud detection and phase determination using SEVIRI aboard the geostationary satellite Meteosat Second Generation. ProPS distinguishes between clear sky, optically thin ice (TI), optically thick ice (IC), mixed phase (MP), supercooled liquid (SC) and warm liquid (LQ) clouds. The Lidar-Radar cloud product DARDAR is used as a reference and a Bayesian approach is applied to combine the cloud and phase information from different SEVIRI channels and prior knowledge. For the probabilities used in the Bayesian approach, we carefully select SEVIRI channels and their dependencies that are used as conditions in the probabilities in order to optimise the information content of the SEVIRI channels. We implement both a day and a night version of the algorithm, with combinations of SEVIRI channels at wavelengths $0.6, 1.6, 8.7, 10.8$ and $12 \, \mu$m, along with a texture parameter derived from the $10.8 \, \mu$m channel. The result of this Bayesian approach is a probability for each cloud state (clear sky and the various cloud phases) per SEVIRI pixel. This allows us to select the most likely cloud state as the final result. ProPS effectively transfers the advanced cloud and phase detection capabilities of DARDAR to the SEVIRI geostationary imager.

We validate the method using six months of independent collocated DARDAR data. Our findings show that the daytime algorithm successfully detects 93% of clouds and 86% of clear sky pixels. It also shows good performance in accurately classifying cloud phases compared to DARDAR data, with probability of detection (POD) values of 91%, 78%, 52%, 58% and 86% for IC, TI, MP, SC and LQ respectively. Distinguishing between MP and SC poses the greatest challenge in the phase classification, as there is a tendency for MPs to be classified as SC and vice versa. This is expected as SC and MP cloud tops occur in very similar circumstances (e.g. similar latitudes and cloud top temperatures) and can have similar radiative properties if an MP cloud top consists predominantly of liquid droplets. However, it should be emphasized that ProPS is capable of distinguishing between them in more than 50% of the cases. The primary challenge for the night version lies in detecting low LQ clouds, particularly when their temperatures are similar to the surface temperature; the night version of ProPS tends to overestimate the occurrence of these LQ clouds. However, the night version of ProPS performs nearly as well as the daytime version in terms of cloud phase detection. This indicates that ProPS is suitable for studying the complete daily cycle of cloud phases. Nevertheless, the algorithm is expected to perform best for each location during the times of the day corresponding to the overflight periods where the sza and umu values as well as their combinations (during daytime) are covered by the DARDAR dataset. Similarly, the prior information used in the retrieval process is only representative for the specific overflight times.

An advantage of the ProPS method is its ability to assign a certainty to the results: In the validation, we observe that the POD of phase detection consistently increases with certainty, providing a straightforward measure of the reliability of the results.

Thus, ProPS represents a significant advancement in discriminating cloud top phases compared to traditional retrieval methods. This distinction is crucial for studying ice in the atmosphere, understanding mixed-phase cloud properties and investigating the cloud radiative forcing associated with phase transitions. The new method enables the study of microphysical and macrophysical cloud properties of clouds with different phases, in particular MP and SC clouds, which have so far been little investigated from geostationary satellites. The geostationary perspective allows the analysis of the temporal evolution of clouds

with different phases as well as phase transitions. SEVIRI, which has been in operation for almost two decades (2004-2023), provides an extensive data set that can be used effectively in conjunction with this method to make valuable statistical comparisons with climate models. Furthermore, ProPS has the advantage of providing probabilities for each cloud state. This could be a valuable additional parameter for comparison with climate models. In terms of further development of the ProPS method, the algorithm can be extended to other satellites with few modifications using for instance spectral band adjustment factors, as proposed by Piontek et al. (2023), since similar channels as used for ProPS are available in most current operational polar and geostationary satellite passive imagers. The Flexible Combined Imager (FCI) aboard the follow-on satellite of MSG (Meteosat Third Generation – MTG, launched on 13 December 2022 (Durand et al., 2015)) has additional channels in the near infrared available, which contain information on the cloud phase (e.g. the $2.2\,\mu$m or $3.8\,\mu$m channel). However, in order to incorporate and use channels not available to SEVIRI that contain phase information, one first needs to collect a data set of collocated active observations to compute the necessary probabilities. In the future, this could be done with the EarthCARE satellite (Wehr et al., 2023) (planned launch May 2024). Furthermore, working with a Bayesian approach offers an additional advantage: The method can be easily adapted to incorporate input from numerical weather prediction (NWP) models as prior probabilities (as suggested by Mackie et al. (2010)). This modification would allow the use of NWP model-derived probabilities for cloud presence and their respective phases as part of the method's framework. This integration promises to improve the accuracy and reliability of the ProPS method in future applications.

*Code and data availability.* MSG/SEVIRI data are available from the EUMETSAT (European Organisation for the Exploitation of Meteorological Satellites) data center. The auxiliary data are available at ECMWF (European Center for Medium-Range Weather Forecasts). DARDAR data are available from the ICARE Data and Services Center at https://www.icare.univ-lille.fr/ (accessed on 12 January 2023). The collocated data set, computed probabilities and the ProPS algorithm presented in this study are available on request from the corresponding author.

## Appendix A: Examples for probabilities

To provide readers with a visual understanding of the Bayesian probabilities computed using the Kernel Density Estimation (KDE) method, we present additional examples in Fig. A1. The figure showcases the probabilities for specific channel combinations, namely $BTD_{10.8-8.7}$, $BTD_{10.8-12}$, $R_{1.6}$, and $RR_{1.6/0.6}$, given the cloud state $q$ (in different colors). The values for the additional conditions are displayed in the figure for each channel (combination).

## Appendix B: Comparison of ProPS and CLAAS-3

In order to better characterize ProPS, we conduct a comparison to the CM SAF CLoud property dAtAset using SEVIRI - Edition 3 (CLAAS-3) product, which was released in 2022 (Meirink et al., 2022). This new edition of the CLAAS product

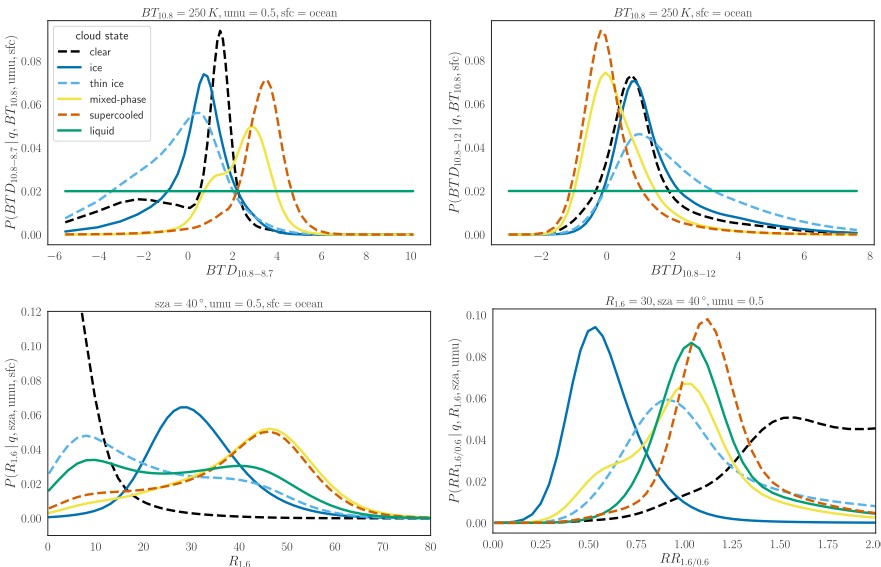

**Figure A1.** Examples for probabilities for different channel (combinations) computed using KDE.

offers an extended phase classification system, distinguishing between clear sky, liquid, supercooled, and various ice cloud
types, which we condensed into one ice cloud category for simplification.

The method of the CLAAS-3 cloud detection, called CMA-prob, is shows some similarities to ProPS, especially because
it uses a Bayesian approach based on the CALIPSO/CALIOP (but not on CloudSat/CPR) cloud mask as ground truth and a
selection of visible and infrared SEVIRI channels as inputs (Karlsson et al., 2017). While this probabilistic methodology is
similar for ProPS and CMA-prob, their tactics differ slightly: CMA-prob does not use conditions (except for surface types)
for the probabilities but instead subtracts pre-calculated image feature thresholds from each channel (combination). These
thresholds are dynamic, depending for instance on satellite geometry and atmospheric conditions. In contrast to ProPS, CMA-
prob assumes independence of the different SEVIRI channel (combinations). Another deviation from ProPS is that CMA-prob
excludes thin ice clouds with optical thickness smaller than 0.2 to prevent overfitting. For the pixels classified as cloudy by
their initial procedure CMA-prob, CLAAS-3 employs a (separated) cloud-top phase determination. It relies on a series of
threshold tests utilizing SEVIRI channels at wavelengths of 3.8, 6.3, 8.7, 10.8, 12.0 and 13.4 $\mu$m as well as clear- and cloudy-
sky simulated IR radiances and brightness temperatures. Additionally, consistency with the cloud optical thickness and particle
effective radius retrieval from solar and NIR channel combinations is demanded (Meirink et al., 2022).

To compare ProPS and CLAAS-3, we use 12 SEVIRI scenes sampled at different seasons and different times of day. Fig-
ure B1 shows one such scene. The circumstances in which ProPS and CLAAS-3 differ in the figure are similar for the other
scenes used in the comparison. Figure B2 is a statistic over all 12 scenes, comparing the classification of CLAAS-3 and ProPS.
Overall, the figures show that there is a good general agreement between the two methods. In Fig. B1, the positions and phases
of the clouds generally agree well when looking at the "big picture". However, there are differences in the details. For cloud

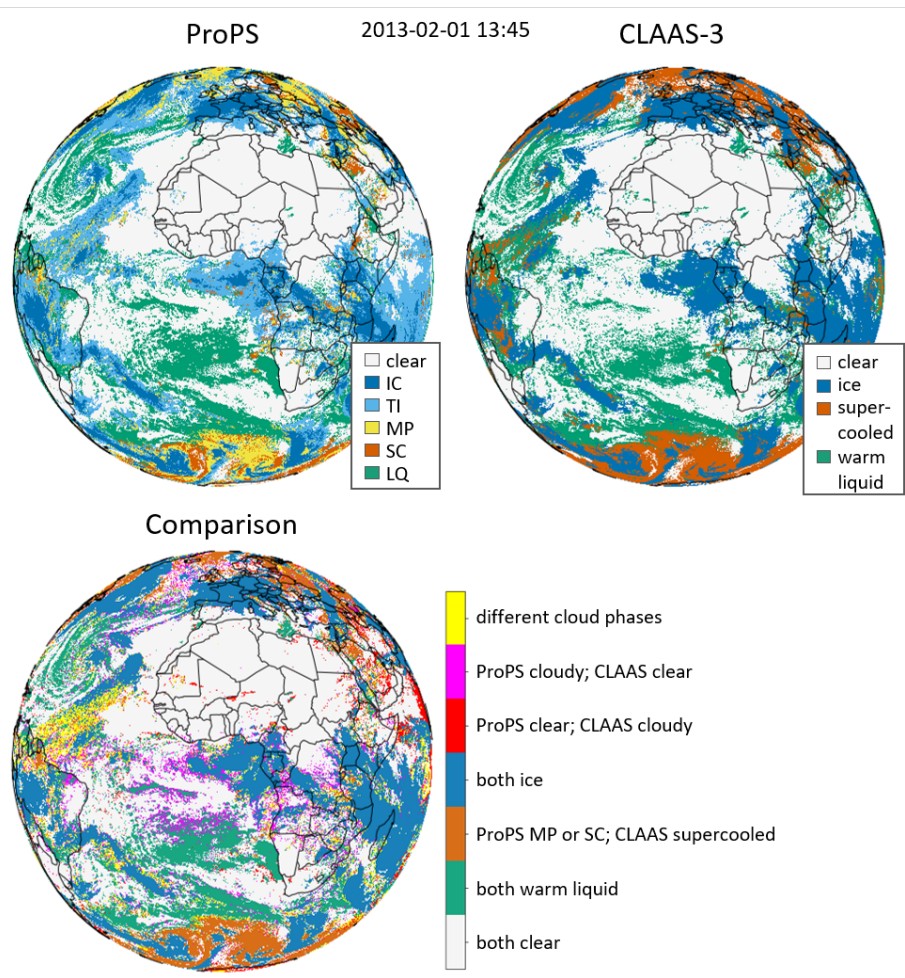

**Figure B1.** Comparison of ProPS with the CM SAF CLoud property dAtAset using SEVIRI - Edition 3 (CLAAS-3) for one exemplary SEVIRI scene. The upper row shows the results of both methods. The lower row shows the comparison of the ProPS and CLAAS-3 results.

detection, discrepancies between ProPS and CLAAS-3 could stem on the one hand from differences in the "training" datasets (ProPS employing DARDAR, while CLAAS-3 utilizes data from CALIPSO). On the other hand, there are some differences
in the selection of SEVIRI channels and the conditions/thresholds employed, as well as the implementation of the Bayesian approach. These nuances likely contribute to the observed differences in cloud and phase detection.

We find that ProPS classifies more pixels as cloudy than CLAAS-3: For the 12 scenes, ProPS classified 62% of all pixels as cloudy, while CLAAS-3 classified 57% as cloudy. The differences between ProPS and CLAAS-3 are often found at the cloud edges, especially for small scale warm cumulus and thin cirrus clouds, both in general difficult cloud types to detect (e.g.
in Fig. B1 the pink areas in the Tropics and the cumulus deck West of Africa). The agreement is better during the day than during the night, as expected. Especially low, warm clouds are difficult to distinguish from the surface using IR channels alone,

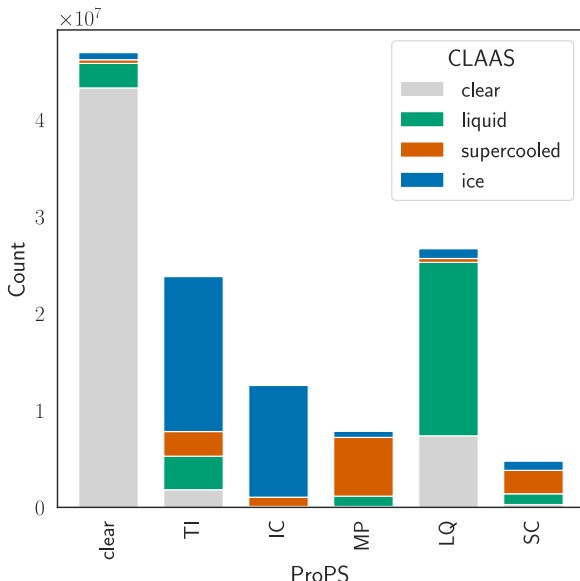

**Figure B2.** Statistic of the comparison of ProPS with CLAAS-3 over 12 SEVIRI scenes sampled at different seasons and different times of day.

leading to the larger discrepancies between ProPS and CLAAS-3 during the night compared to the day: During the day, ProPS and CLAAS-3 agree on the classification of 81% of all pixels; during the night they agree on 78% of all pixels. For thin ice clouds, the difference between the two methods might come (partly) from the exclusion of clouds with OT smaller than 0.2 in

CLAAS-3. In general, ProPS tends to overestimate rather than underestimate the amount of cloud (as discussed in Section 6), i.e. it is a clear sky conservative algorithm, whereas CLAAS-3 seems to be a cloud conservative algorithm. Exceptions are high satellite zenith angles (> 70°) and bright surfaces (deserts, ice, snow), where CLAAS-3 has higher cloudiness values compared to ProPS.

     Next, we take a look at the phase categorization of both methods. ProPS has an additional phase category, namely MP,

which has no direct correspondence in CLAAS-3. We find that clouds classified as MP by ProPS are mostly categorized as supercooled by CLAAS-3; almost no ProPS MP clouds are classified as ice by CLAAS-3. The CLAAS-3 supercooled clouds are also the largest contribution to the ProPS SC category. The main differences in phase detection (as the cloud detection) are found at cloud edges or at the transition regions between different phases (in Fig. B1 for instance at the transition between supercooled and warm liquid clouds over the Southern Ocean). The phase category of ProPS which differs the most from

CLAAS-3 are thin ice clouds (see TI bar in Fig. B2): ProPS categorizes more pixels as thin ice than CLAAS-3. In most cases, ProPS and CLAAS-3 agree on the existence and position of thin ice clouds, however they often have a larger extent in ProPS (see the yellow regions in Fig. B1 at ice cloud edges). These differences might be due to the mentioned exclusion of clouds with OT smaller than 0.2 in CLAAS-3. The high sensitivity of ProPS to thin ice might however also lead to false alarms. CLAAS-3

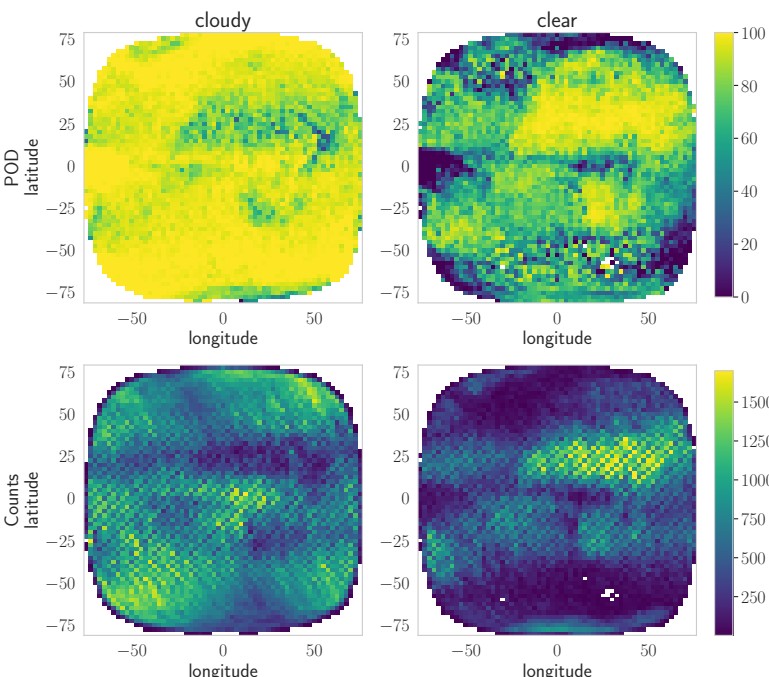

**Figure C1.** As Fig. 10, but for the ProPS nighttime version.

categorises parts of the SC and MP categories of ProPS as warm liquid (green parts of MP and SC bars in Fig. B2), suggesting
a tendency towards warmer cloud types in the CLAAS-3 classification scheme compared to ProPS.

## Appendix C:  ProPS night version performance on the SEVIRI disc

In Fig. C1 and Fig. C2 we show the POD of the night version of ProPS on the SEVIRI disc for the six months of validation data,
for cloud detection and phase detection (given a detected cloud) respectively. The upper panels show the POD of each cloud
state and the lower panels show the corresponding distribution of the number of occurrences of each cloud state according to
DARDAR. The figures show that the POD of clear sky is worse in the night time version almost everywhere in the SEVIRI
disc except for the desert regions on the African continent. The POD of clouds on the other hand is similar to the day time
version, suggesting that ProPS has a tendency to overestimate cloudiness during the night. The distribution of the POD of the
different phases is very similar to the daytime version.

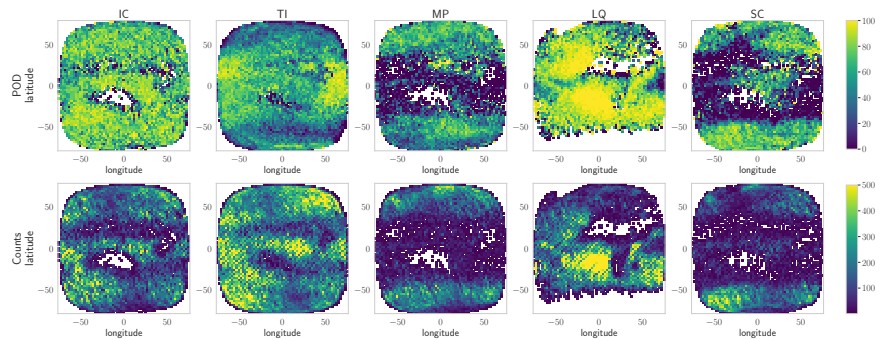

**Figure C2.** As Fig. 11, but for the ProPS nighttime version

*Author contributions.* All authors contributed to the project through discussions. JM and LB conceived the concept of this study. JM developed the presented methods and carried out the analysis with help from LB and valuable feedback from BM. JM and DP implemented the algorithm for the retrieval. CV supervised the project and provided scientific feedback. JM took the lead in writing the manuscript. All authors provided feedback on the manuscript.

*Competing interests.* The authors declare that they have no conflict of interest.

*Acknowledgements.* We thank F. Ewald for constructive discussions and valuable feedback. This research was funded by the Deutsche Forschungsgemeinschaft (DFG, German Research Foundation)–TRR 301–Project-ID 428312742.

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
