# Peer review of "Bayesian Cloud Top Phase Determination for Meteosat Second Generation"

_EGUsphere, 2023_

## Referee Comment (RC1)

**Review of the paper "Bayesian Cloud Top Phase Determination for Meteosat Second Generation" by Mayer et al.**

The manuscript is accurate, well-structured and well written. I recommend for it to be accepted after the authors have addressed the few minor points I list below.

**General comments**

One thing that is mentioned in the abstract and in the introduction is the low/inadequate accuracy of traditional cloud typing methods in detecting cloud top phase. Perhaps a comparison with at least one another product available online for cloud type retrieval (could be from geostationary or low earth orbit instruments) could be interesting. It could be briefly shown as direct comparison or discussed. It could be for example the NWC-SAF Cloud Type product (even though it does not explicitly report the cloud phase, though the cloud types can be broadly related to cloud phase).

In the discussion in chapter 8.1 it is mentioned that a difficult case is represented, perhaps unsurprisingly, by overlapping cloud layers with a high thin ice cloud over a low liquid cloud. Does the measure of certainty provide an extra information that can be used to isolate these cases? More in general, it would be interesting also to show an example of how the certainty measure looks like for a typical example of the product as in Fig. 7. This is very useful especially when the first phase choice is only marginally more certain than the second choice (as it is mentioned in chapter 8.2 in relation to the POD of MP and SC types). This discussion could be added to chapter 8.3 which currently is fairly limited in content.

A clarification regarding the certainty measure: how should it be interpreted when the probabilities of the two most probable states are both about 0.5 (one slightly higher than the other), while for all the other states probabilities are ~0? In this case has the certainty measure the same meaning as for a case where for example $P(q*|M,A)=0.5$ and the remaining 5 states have all $P=0.1$?

**Minor and technical comments**

Abstract, line3: "mainly distinguished between" -> mainly identified/detected

Section 3.1, line 124: the sentence "These probabilities…" is repeated twice.

Section 4.1: not clear where the information content for the prior is shown in Fig 3. I imagine is the first panel, but from the figure caption is not clear.

Section 4.3, Line 229: this applies also to fractional cloud cover?

Section4.3: given the strong dependency on the surface emissivity at wavelength around 8.7 $\mu$m, should an emissivity map also be taken into account, or the surface type is enough to account for the effect?

Section 4.5: The surface type can be used as a proxy for surface albedo, but does this also include the spectral variation of the albedo? Are significant changes in surface albedo between 0.6 $\mu$m and 1.6 $\mu$m (e.g. over snowy surfaces) important in this context or the surface type is enough?

Section 4.6: why the mutual information between $RR_{1.6/0.6}$ and q seems to benefit by the inclusion of the surface type or the $R_{1.6}$ but not from the use of both together?

Section 8.1, line 388: "by nature/DARDAR to it" not very clear, please rephrase. Also, missing bracket in "(see Fig 7".

Section 8.1, line 418: "spatially shifted against the DARDAR clouds" please clarify.

Section 8.2, line 438: "less sensitive to optically thin clouds that->than Lidar"

Section 8.2, line 480: As discussed in the text, is it clear that SC and MP types are often difficult to distinguish, and that the certainty computed by ProPS is often marginally higher for either of the two types. Also, what is the confidence in the DARDAR supercooled water classification?

Section 9: It would be interesting to briefly discuss how a new sensor (e.g. MTG FCI) would impact such an algorithm, both in terms of the information contained in the new channels and in the surface resolution.

Figure 3: the caption could be clearer. Also, at first I did not understand that the first column of each panel (apart from the third) represents the starting point of each information content before the introduction of each new condition.

Figure 6: perhaps adding another RGB composite helps a better comparison with the categorization as many of the high thin cirrus are lost in the RGB shown in the current figure.

---

## Author Comment (AC1)

Dear Reviewer,

thank you for taking the time to review our manuscript. Your feedback is greatly appreciated and was helpful in improving the quality of this research. We value your constructive criticism and thoughtful comments, which have helped to identify areas that require further clarification and refinement.

We carefully considered your suggestions and incorporated them into the revised manuscript to address the issues raised, as specified below (referee comments in blue; our answers in black).

**General comments**

One thing that is mentioned in the abstract and in the introduction is the low/inadequate accuracy of traditional cloud typing methods in detecting cloud top phase. Perhaps a comparison with at least one another product available online for cloud type retrieval (could be from geostationary or low earth orbit instruments) could be interesting. It could be briefly shown as direct comparison or discussed. It could be for example the NWC-SAF Cloud Type product (even though it does not explicitly report the cloud phase, though the cloud types can be broadly related to cloud phase).

Comparing the ProPS method with another established cloud phase detection product is a very valuable suggestion. We have taken your advice and made a thorough comparison between ProPS and the CM SAF CLoud property dAtAset using SEVIRI - Edition 3 (CLAAS-3) product. We have included the main results of this comparison in the paper and added a detailed discussion to the appendix of the paper, as follows:

[revised manuscript text omitted]

In the discussion in chapter 8.1 it is mentioned that a difficult case is represented, perhaps unsurprisingly, by overlapping cloud layers with a high thin ice cloud over a low liquid cloud. Does the measure of certainty provide an extra information that can be used to isolate these cases? More in general, it would be interesting also to show an example of how the certainty measure looks like for a typical example of the product as in Fig. 7. This is very useful especially when the first phase choice is only marginally more certain than the second choice (as it is mentioned in chapter 8.2 in relation to the POD of MP and SC types). This discussion could be added to chapter 8.3 which currently is fairly limited in content.

We agree that a more detailed discussion of the certainty measure and its meaning is valuable. We have therefore added a certainty measure panel to the example curtains in Fig. 7 as suggested (see figure below). We have added a brief discussion of the meaning of the certainty measure in the example figures in the discussion of challenging situations for the ProPS retrieval in section 8.1 as follows:

[revised manuscript text omitted]

A clarification regarding the certainty measure: how should it be interpreted when the probabilities of the two most probable states are both about 0.5 (one slightly higher than the other), while for all the other states probabilities are ~0? In this case has the certainty measure the same meaning as for a case where for example $P(q^*|M,A)=0.5$ and the remaining 5 states have all $P=0.1$?

As this comment correctly describes, the certainty measure cannot capture all the information contained in the posterior $P(q \mid M,A)$: Compare for instance the MP and SC clouds approximately in the middle of both scenes in Fig.7. In 7(a), only probabilities for MP and SC are above zero, meaning that the algorithm is fairly certain that it is one or the other. In 7(b) also TI and IC have probabilities above zero, which mirrors the more complicated cloud scene in this case where the MP and SC clouds are sometimes interrupted by ice in between. If both cases have similar certainty values, one would need to look at the posterior probabilities to get more detailed information.

**Minor and technical comments**

Abstract, line3: "mainly distinguished between" -> mainly identified/detected

We changed the phrase to "mainly detected".

Section 3.1, line 124: the sentence "These probabilities…" is repeated twice.

Thank you for noticing this mistake, we deleted it.

Section 4.1: not clear where the information content for the prior is shown in Fig 3. I imagine is the first panel, but from the figure caption is not clear.

We agree that the caption was unclear in this point. We rewrote the caption to explain Fig. 3 in more detail:

"First panel: Mutual information $I$ between the latitude and the cloud state q (first row), cloudy/clear, abbreviated as c/c, (second row) and cloud phases (third row) for different sets of conditions $C$. This represents the information content of the different priors we considered, where latitude is a fixed condition, i.e. P (q | lat, $C$). Other panels: Mutual information $I$ between SEVIRI channel (combinations) and cloud state q, cloudy/clear and cloud phases for different sets of conditions $C$. Empty spaces for $C$ mean no condition, i.e. the starting point of $I$ before conditions are introduced. The different mutual information values for q, cloudy/clear and phase indicate whether a channel (combination) contributes more to cloud or phase detection. The blue boxes indicate the

sets of conditions selected for ProPS."

**Section 4.3, Line 229: this applies also to fractional cloud cover?**

Yes, that is correct. We added this aspect to the sentence.

**Section4.3: given the strong dependency on the surface emissivity at wavelength around 8.7 μm, should an emissivity map also be taken into account, or the surface type is enough to account for the effect?**

We agree that an emissivity map could provide additional information for the BTD (10.8 - 8.7). However, we believe that the surface type constraint covers most of the emissivity differences for the channels. The advantages of the surface type variable are that 1) it can be used in several of the probabilities and only needs to be read in once, 2) it is (most of the time) a temporally fixed quantity that does not need to be retrieved and regridded for each time step, thus saving computational cost, and 3) it is a categorical variable rather than a continuous variable, which means that the data requirements for calculating probabilities using surface type as a condition are less than for using emissivity. For these practical reasons we chose surface type as a condition for the probability.

**Section 4.5: The surface type can be used as a proxy for surface albedo, but does this also include the spectral variation of the albedo? Are significant changes in surface albedo between 0.6 μm and 1.6 μm (e.g. over snowy surfaces) important in this context or the surface type is enough?**

Since the probabilities for the reflectance ratio are computed from the measured DARDAR data conditioned on the surface type, they already implicitly include the information on the spectral variation of the albedo for the given surface type. Only when the spectral variation of the albedo changes between pixels of the same surface type this is not included in the computed probabilities. However, we believe that in most situations this effect plays a minor role compared to the other conditions used. For this reason, and for practical reasons (mainly the limited amount of DARDAR 'training' data available), we have chosen not to include albedo.

**Section 4.6: why the mutual information between RR1.6/0.6 and q seems to benefit by the inclusion of the surface type or the R1.6 but not from the use of both together?**

This is an interesting question, but it is not easy to answer. One would need to study the information content of the channels and the different conditions in more detail, which is beyond the scope of this paper. In general, however, the mutual information does not have to increase when more conditions are added. For example, if we condition the mutual information between variables X and Y on a parameter C that is a confounder or mediator of X and Y, the mutual information will typically decrease (if there are no other parameters that lead to opposite effects).

**Section 8.1, line 388: "by nature/DARDAR to it" not very clear, please rephrase. Also, missing bracket in "(see Fig 7".**

We made the statement clearer by rephrasing to "…highlight the strengths of the ProPS retrieval and the challenges posed by, for example, complex cloud scenes or the different viewing geometries of polar orbiting and geostationary satellites"

Section 8.1, line 418: "spatially shifted against the DARDAR clouds" please clarify.

To make this clearer, we have added the following sentence, referring to examples in Fig. 7: "Often, the ProPS $q_*$ is spatially slightly shifted against the DARDAR results, especially in the high latitude example in Fig. 7(a) where $q_*$ is often slightly shifted to the left relative to $q_{dardar}$."

Section 8.2, line 438: "less sensitive to optically thin clouds that->than Lidar"
Thank you for noticing this mistake, we corrected it.

Section 8.2, line 480: As discussed in the text, is it clear that SC and MP types are often difficult to distinguish, and that the certainty computed by ProPS is often marginally higher for either of the two types. Also, what is the confidence in the DARDAR supercooled water classification?

As the referee correctly describes, MP and SC are the two cloud states which are most difficult to distinguish. This can be seen from the example curtains in Fig. 7, from the POD of ProPS in Fig. 8 and from the on average lower certainty values of MP and SC compared to other cloud states (Fig. 9).

Regarding the confidence of DARDAR in supercooled water classification, unfortunately there is no uncertainty of the retrieved classification of DARDAR available in the product. In general, the phase classification in DARDAR is done using temperature information from ECMWF, the lidar backscatter, radar reflectivity and cloud layer thickness as criteria (Ceccaldi et al., 2013): Cloud layers containing supercooled water are identified by their strong lidar backscatter and subsequent attenuation in temperature ranges between 0 ∘C and -40 ∘C. A further distinction into pure supercooled water without ice crystals is made using the absence of radar return, since the diameter of cloud droplets is mostly below the CloudSat sensitivity (Hogan et al., 2003). If the layer is thicker than 300 m in the temperature range 0 ∘C to -40 ∘C, it is assumed to be fully glaciated.

Section 9: It would be interesting to briefly discuss how a new sensor (e.g. MTG FCI) would impact such an algorithm, both in terms of the information contained in the new channels and in the surface resolution.

We agree that the application of ProPS to new sensors is an interesting point. We therefore extended the discussion on further developments, focusing on the initial difficulty that new channels would need collocated active data in order to use them:

"In terms of further development of the ProPS method, the algorithm can be extended to other satellites with few modifications using for instance spectral band adjustment factors, as proposed by Piontek et al. (2023), since similar channels as used for ProPS are available in most current operational polar and geostationary satellite passive imagers. However, in order to incorporate and use channels not available to SEVIRI that contain phase information, such as additional channels in the near infrared of the Flexible Combined Imager (FCI) aboard the follow-on satellite of MSG (Meteosat Third Generation – MTG, launched on 13 December 2022 (Durand et al., 2015)), one first needs to collect a data set of collocated active observations to compute the necessary probabilities. In the future, this could be done with the EarthCARE satellite (Wehr et al., 2023) (planned launch May 2024)."

Figure 3: the caption could be clearer. Also, at first I did not understand that the first column of each panel (apart from the third) represents the starting point of each information content before the introduction of each new condition.

We have rewritten the caption of Figure 3 to clarify the unclear points (see above in response to the comment on Section 4.1).

Figure 6: perhaps adding another RGB composite helps a better comparison with the categorization as many of the high thin cirrus are lost in the RGB shown in the current figure.

Unfortunately, in many of the RGB composites in which thin ice are better visible, the low clouds are not very well visible. We therefore decided to keep the "natural color" RGB composite as a compromise for visibility of both high and low clouds.

---

## Author Comment (AC2)

Dear Reviewer,

thank you for taking the time to review our manuscript. Your feedback is greatly appreciated and was helpful in improving the quality of this research. We value your constructive criticism and thoughtful comments, which have helped to identify areas that require further clarification and refinement.

The topic has significant interest for both weather and climate applications. The paper is well written and organized. One suggestion is some more detailed discussions on the ground truth data (DARDAR) could be added for limitations and further improvements. A few minor revisions below, mostly for more clarification, will also help to improve the manuscript for publication.

We carefully considered your suggestions and incorporated them into the revised manuscript to address the issues raised, as specified below (referee comments in blue; our answers in black).

We have added more details about the DARDAR data, mainly in section 2.1. These additions are outlined in the responses to the referee's comments below (see in particular the responses to comments on lines 46-47 and 76-77 above).

Line 5 Abstract: Add (PRObabilistic cloud top Phase retrieval for Seviri) to ProPS

We added the suggestion.

Line 30-34: Add full name of GOES: Instead of GOES-R/S, GOES-R series or GOES-16/17/18: Add sensors: ABI and AHI with references, like SEVIRI. GOES -> GOES-R

We added the names of the imagers and corrected the GOES-R name.

Line 40:" the Lidar-Radar cloud product DARDAR…"  Add a brief summary for readers who are not familiar with this data, space-borne data derived from CloudSat-CALIPSO, even though the details are followed in the next section but it appears first here.

We added a brief description of DARDAR as follows:

"We use the Lidar-Radar cloud product DARDAR (liDAR/raDAR, Delanoë and Hogan, 2010) as the basis for this method. DARDAR is based on the combination of active radar and lidar measurements from the A-Train satellites CloudSat and CALIPSO and provides a consolidated classification of the measured clouds into different cloud phases."

Line 46-47: For "DARDAR as ground truth" - If the data is temperature only-based, still limitations especially for supercooled and mixed? If any, it would be better to include some discussions on the of the ground truth in the data or conclusion sections.

The DARDAR phase classification is not only based on temperature, but also uses the lidar backscatter, radar reflectivity and cloud layer thickness as criteria. DARDAR combines the sensitivity of lidar to optically thin cirrus with the ability of radar to penetrate optically thicker clouds in the following way (explanation following Mayer et al. (2023)): Atmospheric targets are labelled as warm liquid clouds where the wet bulb temperature is > 0 °C, calculated from temperature, pressure and humidity from the ECMWF-AUX dataset (Benedetti, 2005). In addition, cloud layers containing supercooled water are identified by their strong lidar backscatter and subsequent attenuation in temperature ranges between 0 ∘C and -40 ∘C. A further distinction into pure supercooled water without ice crystals is made using the absence of radar return, since the diameter of cloud droplets is mostly below the CloudSat sensitivity (Hogan et al., 2003). If the layer is thicker than 300 m in the

temperature range 0 ◦C to -40 ◦C, it is assumed to be fully glaciated. Further details can be found in Ceccaldi et al. (2013).

We extended the explanation in the paper to make these points clear as follows:

"As ground truth for cloud occurrence and cloud thermodynamic phase, this study uses the product DARDAR-MASK, part of the synergistic active remote sensing product DARDAR (Delanoë and Hogan, 2010; Ceccaldi et al., 2013). DARDAR-MASK is derived from the sun-synchronous, low-earth orbit satellites CloudSat (Stephens et al., 2002) and CALIPSO (Winker et al., 2003). To distinguish between cloud phases, DARDAR-MASK uses the wet bulb temperature derived from the ECMWF-AUX dataset (Benedetti, 2005) and the extent of cloud layers as well as the different sensitivities of lidar and radar to cloud particles of varying sizes: cloud layers containing water have a strong lidar backscatter and subsequent attenuation; the CloudSat radar is mostly only sensitive to the larger ice crystals (Hogan et al., 2003). DARDAR-MASK provides vertically resolved cloud thermodynamic phase along the track of the CALIPSO and CloudSat satellites with a spatial resolution of 1.1 km along track and 60 m in the vertical direction."

Line 76-77 Add more details. Did the authors use the cloud top phase info in DARDAR from CloudSat-CALIPSO as is? Assumed there was no further consideration on the lower layer phase from the active-sensors for passive radiometers like SEVIRI, correct? What exactly mixed phase is defined?

To answer these questions and make the collocation procedure clearer, we extended the explanation of the collocation of DARDAR and SEVIRI in Sect. 2.1 as follows:

"From the DARDAR data we extract two key pieces of information for each SEVIRI pixel: 1) whether a pixel is clear or cloudy, and 2) a cloud top phase. This cloud top phase at SEVIRI resolution is defined by horizontal and vertical averaging of DARDAR gates using a simplified penetration depth (Mayer et al., 2023). We distinguish between warm liquid (LQ), supercooled liquid (SC), mixed phase (MP) and ice. MP cloud tops in SEVIRI resolution are defined as containing either only gates classified as mixed-phase by DARDAR or a mixture of liquid, ice and/or mixed-phase DARDAR gates in the cloud top gates considered for the collocation (see Mayer et al. (2023) for details). To ensure that the averaging over DARDAR gates for a SEVIRI pixel is not done over two different clouds, the gates are all required to have a similar cloud top height. For multilayered clouds, e.g., a high cirrus cloud on top of lower clouds, only the uppermost cloud layer is considered."

Line 54 and nighttime eval.in sect 8: Have you ever considered 3.9 um or a channel difference including this, particularly for nighttime retrievals?

We have considered the 3.9 $\mu$m channel, but decided against using it, since CO2 and water vapor absorption effects can have a possibly large influence on the channel, which we can not account for. (see for instance https://www-cdn.eumetsat.int/files/2020-04/pdf_conf_p50_s10_05_charvat_p.pdf)

Line 65: "DARDAR (liDAR/raDAR, Delanoë and Hogan, 2010)" repeated in Intro and here.

Thank you for noticing; we deleted it.

Line 66: Probably need references for these satellites, CloudSat and CALIPSO, although they are well know.

We added the corresponding citations.

Line 72: MET-9 -> maybe better to write the full name, Meteosat-9?

We rewrote to: "SEVIRI aboard the geostationary satellite Meteosat-9 (part of the Meteosat Second Generation series)".

Line 79: "…observed (see Mayer et al. (2023) for details)." - more info about the phase data used as a ground truth in this study will be desirable here, briefly from Mayer et al. (2023) if needed.

We added more details about the "ground truth" DARDAR data as well as the collocation procedure with SEVIRI in Sect. 2.1 (see the responses to comments on lines 46-47 and 76-77 above).

Line 98: "combinations, at probability distributions are used" – more clarification?

We rewrote the passage to make it clearer as follows:

"For the regions of the parameter space without samples for high sza and umu combinations, the solar channels are effectively not used. In a Bayesian update, this is done by imposing flat probability distributions for the solar channels in these regions of the parameter space, i.e. the cloud state probabilities are not changed by the solar channels.

The Bayesian update is further explained in Sect. 6.

Figure 2: Hope the figure quality with larger fonts can be added in the final version.

We increased the figure resolution and quality for the final version.

Line 135: M2 means the other channel measurement?

Yes, that is correct. To make that clearer in the text, we added "SEVIRI" in the sentence:

"Updating the probability with a second SEVIRI measurement M2 leads to…"

Line 157: "most likely": Any minimum threshold which may give 'uncertain'?

No, we always use the most likely cloud state, i.e. the cloud state with the highest probability for each SEVIRI pixel. However, to quantify the uncertainty of a retrieved result, we define a "certainty parameter" from the computed probabilities (explained in Sect. 3.3).

Eq 7: "season" - How was it quantified in the computation?

We defined "season" in Eq. 7 more clearly as "… and season is one of the four seasons of the year (December - January - February, March - April - May, June - July - August or September - October - November).".

Line 201: "parameters A introduced above in Sect. 7 " -> need to be corrected?

Thank you for noticing this mistake, we corrected it.

Line 203-205: Not very clear. Could you explain more details? It is explaining Fig. 3, right?

Yes, it refers to Fig. 3, which shows the mutual information values of the prior (first panel) for different sets of conditions. We expanded the caption of Fig. 3 to make it clearer (see two questions below). We also rewrote the explanation in the text as follows:

"Furthermore, our mutual information calculations show that conditioning on latitude, longitude and season yields the prior with the optimal information content compared to other possible sets of conditions (see Fig. 3). This means that location (latitude and longitude) and season are the main dependencies."

Line 207: BT: - What about putting this acronym to the place where it appears first, and using BT for consistency?

Good point, we changed it accordingly.

Figure 3: Higher resolution one with bigger fonts, please. The figure caption doesn't seem very straightforward, please more clarify it for better understanding?

We increased both the resolution and the font. We extended the caption to make it clearer as follows:

"First panel: Mutual information $I$ between the latitude and the cloud state q (first row), cloudy/clear, abbreviated as c/c, (second row) and cloud phases (third row) for different sets of conditions $C$. This represents the information content of the different priors we considered, where latitude is a fixed condition, i.e. P(q | lat, $C$). Other panels: Mutual information $I$ between SEVIRI channel (combinations) and cloud state q, cloudy/clear and cloud phases for different sets of conditions $C$. Empty spaces for $C$ mean no condition, i.e. the starting point of $I$ before conditions are introduced. The different mutual information values for q, cloudy/clear and phase indicate whether a channel (combination) contributes more to cloud or phase detection. The blue boxes indicate the sets of conditions selected for ProPS."

Line 214: "the brightness temperature" -> BT10.8

We changed brightness temperature to BT10.8.

Line 217: BT -> BT10.8

We changed it accordingly.

Line 276: Bayesian retrieval methods -> specify which cloud property retrievals

We specified in the text that texture parameters have been used in Bayesian retrieval methods for cloud detection.

Line 376: Fig. 6 -> Figure 6

Thank you for noticing this; we changed it.

Line 379: "...retrieval detects (most) clouds which..." Can we think the retrieval method is for cloud detection in the first place, not just cloud phase discrimination?

Yes, that is correct. The method combines a cloud detection and phase discrimination since it distinguishes between clear sky and the different cloud types (TI, IC, MP, SC, LQ).

Line 379: Add the full name for ITCZ, even though most of us know what it is.

We added the full name for ITCZ.

Line 523-524: Again, addition to using 3.9 um info, have you ever considered additional environment parameters from ERA-5 such as low level humidity and SST/low level atmos temperature combinations?

We considered using additional parameters from ERA-5. However, for this first version of ProPS we decided not to do so for practical reasons: 1) The data requirements to compute statistically valid probabilities scale with the number of parameters used. We therefore limited the number of parameters for each probability. 2) Parameters from ERA-5 have to be retrieved and regridded for each SEVIRI time step, which increases the computational cost.

However, the inclusion of more parameters from ERA-5 could be reconsidered in future developments of the method.

Appendix B: I would think this discussion can be part of the main section 8.

We added a brief discussion of the results of the ProPS night version in the SEVIRI disc to section 8.4:

"For the night version of ProPS, the POD of clouds is similar to the day version while the POD of clear sky is slightly lower almost everywhere in the SEVIRI disc (see Fig. C1). This suggests that ProPS tends to overestimate cloudiness during the night. The spatial distribution of the POD of the different phases is very similar to the daytime version (see Fig. C2)."

References: Cover 1999: Some info still missed.

Thank you for noticing. We added the missing information in the reference.

**References**

Benedetti, A.: CloudSat AN-ECMWF ancillary data interface control document, technical document, CloudSat Data Processing Cent., Fort-Collins, Colo., (Available at http://cloudsat.cira.colostate.edu/ICD/AN-ECMWF/AN-ECMWF_doc_v4.pdf), 2005.

Ceccaldi, M., Delanoë, J., Hogan, R. J., Pounder, N. L., Protat, A., and Pelon, J.: From CloudSat-CALIPSO to EarthCare: Evolution of the DARDAR cloud classification and its comparison to airborne radar-lidar observations, Journal of Geophysical Research: Atmospheres, 118, 7962–7981, https://doi.org/10.1002/jgrd.50579, 2013

Delanoë, J. and Hogan, R. J.: Combined CloudSat-CALIPSO-MODIS retrievals of the properties of ice clouds, Journal of Geophysical Research, 115, https://doi.org/10.1029/2009JD012346, 2010

Hogan, R. J., Francis, P. N., Flentje, H., Illingworth, A. J., Quante, M., and Pelon, J.: Characteristics of mixed-phase clouds. I: Lidar, radar and aircraft observations from CLARE'98, Quarterly Journal of the Royal Meteorological Society, 129, 2089–2116, https://doi.org/10.1256/rj.01.208, 2003

Mayer, J., Ewald, F., Bugliaro, L., and Voigt, C.: Cloud Top Thermodynamic Phase from Synergistic Lidar-Radar Cloud Products from Polar Orbiting Satellites: Implications for Observations from Geostationary Satellites, Remote Sensing, 15, 1742, https://doi.org/10.3390/rs15071742, 2023.

Stephens, G. L., Vane, D. G., Boain, R. J., Mace, G. G., Sassen, K., Wang, Z., Illingworth, A. J., O'connor, E. J., Rossow, W. B., Durden, S. L., Miller, S. D., Austin, R. T., Benedetti, A., and Mitrescu, C.: THE CLOUDSAT MISSION AND THE A-TRAIN: A New Dimension of Space-Based Observations of Clouds and Precipitation, Bulletin of the American Meteorological Society, 83, 1771–1790, https://doi.org/10.1175/BAMS-83-12-1771, 2002

Winker, D. M., Pelon, J. R., and McCormick, M. P.: The CALIPSO mission: spaceborne lidar for observation of aerosols and clouds, https://doi.org/10.1117/12.466539, 2003